# Examining the evidence for decoupling between photosynthesis and transpiration during heat extremes

Martin G. De Kauwe[1], Belinda E. Medlyn[2], Andrew J. Pitman[1], John E. Drake[3], Anna Ukkola[4], Anne Griebel[2], Elise Pendall[2], Suzanne Prober[5] and Michael Roderick[4]

[1]ARC Centre of Excellence for Climate Extremes and the Climate Change Research Centre, University of New South Wales, Sydney, NSW 2052, Australia
[2]Hawkesbury Institute for the Environment, Western Sydney University, Locked Bag 1797, Penrith NSW 2751 Australia
[3]Forest and Natural Resources Management, SUNY-ESF, Syracuse, NY, USA.
[4]ARC Centre of Excellence for Climate Extremes and the Research School of Earth Sciences, Australian National University, Canberra, ACT, 2601, Australia.
[5]CSIRO Ecosystem Sciences, Private Bag 5, Wembley, Western Australia 6913, Australia

*Correspondence to*: Martin G. De Kauwe (mdekauwe@gmail.com)

**Abstract.** Recent experimental evidence suggests that during heat extremes, wooded ecosystems may decouple photosynthesis and transpiration, reducing photosynthesis to near zero but increasing transpiration into the boundary layer. This feedback may act to dampen, rather than amplify, heat extremes in wooded ecosystems. We examined eddy-covariance databases (OzFlux and FLUXNET2015) to identify whether there was field-based evidence to support these experimental findings. We focused on two types of heat extremes: (i) the three days leading up to a temperature extreme, defined as including a daily maximum temperature > 37°C (similar to the widely used TXx metric) and (ii) heatwaves, defined as three or more consecutive days above 35°C. When focussing on (i), we found some evidence of reduced photosynthesis and sustained or increased latent heat fluxes in seven Australian evergreen wooded flux sites. However, when considering the role of vapour pressure deficit and focusing on (ii), we were unable to conclusively disentangle the decoupling between photosynthesis and latent heat flux from the effect of increasing vapour pressure deficit. Outside of Australia, the Tier-1 FLUXNET2015 database provided limited scope to tackle this issue as it does not sample sufficient high temperature events with which to probe the physiological response of trees to extreme heat. Thus, further work is required to determine whether this photosynthetic decoupling occurs widely, ideally by matching experimental species with those found at eddy-covariance tower sites. Such measurements would allow this decoupling mechanism to be probed experimentally and at the ecosystem scale. Transpiration during heatwaves remains a key issue to resolve, as no land surface model includes a decoupling mechanism, and any potential dampening of the land-atmosphere amplification is thus not included in climate model projections.

## 1 Introduction

In response to a warming climate, heatwaves have increased in frequency, magnitude and duration (Alexander et al., 2006; Perkins et al., 2012). Coupled climate models from the Coupled Model Intercomparison Project (CMIP5) project a marked increase in the frequency and severity of these heat extremes (Coumou and Robinson, 2013, Sillmann et al. 2013), highlighting the urgent need to understand the underlying driving mechanisms. Whilst heatwaves are commonly associated with large-scale, high-pressure synoptic systems (anticyclones) (Perkins, 2015), there is increasing evidence of the role of the land-surface in the amplification of heat extremes (Fischer et al., 2007; Teuling et al., 2010; Miralles et al., 2012; Kala et al., 2016; Donat et al. 2017). This land-atmosphere feedback is driven by drying soils and an increase in the sensible heat flux which further warms the boundary layer (Lorenz et al., 2010; Seneviratne et al., 2006). The combination of heat advection and heat storage in the boundary layer is recycled back to the surface over successive days and can lead to increasingly intense heatwaves, including "mega-heatwaves" (Miralles et al., 2014).

A number of studies have highlighted the contrasting functional traits of grasslands and forests as important controls on the role of the land surface in the amplification of heatwaves (Teuling et al. 2010; van Heerwaarden and Teuling, 2014). Grasses often have shallow root profiles, meaning that a relatively small reduction in soil moisture can stress a grassland, resulting in decreased transpiration (either directly via reduced stomatal conductance and/or indirectly via reduced leaf area), leading to a repartitioning of the available (radiant) energy towards sensible heat. Heatwaves also affect forests, but the deeper root profiles that characterise forests may make surface drying less likely to influence the surface energy balance. However, whilst this slower soil water depletion may buffer the transition to increased sensible heat flux, ultimately the decline in soil moisture may still result in heatwave intensification during prolonged dry spells (Teuling et al., 2010).

On the other hand, recent experimental evidence has highlighted a previously overlooked vegetation-atmosphere feedback that may in fact dampen, rather than amplify, heat extremes. A number of heatwave experiments carried out in well-watered, potted plants, have suggested that during temperature extremes, photosynthesis and stomatal conductance ($g_s$) become decoupled, such that photosynthesis is reduced to near zero, but transpiration is maintained (Ameye et al. 2012; von Caemmerer and Evans, 2015; Urban et al. 2017). For example, in a growth chamber study, Urban et al. (2017) found that $g_s$ increased with rising temperature despite photosynthetic activity shutting down for both *Pinus taeda* and *Populus deltoides* x *nigra*. This result was also confirmed in a field-based whole tree-chamber study by Drake et al. (2018), who reported that transpiration was increased and decoupled from photosynthesis in 6-m tall *Eucalyptus parramattensis* trees during an imposed heatwave of four consecutive days with temperatures exceeding 43°C. Crucially, in the Drake et al. (2018) study, the plants were not well-watered. Instead, these trees had been subject to an imposed one-month drought prior to the experiment to reduce soil water

stores. Evidence that transpiration increases during a heatwave, resulting in a cooler canopy temperature, would be consistent with an active mechanism (Trewavas et al. 2009) by trees to cool their canopies. Such a response to heat extremes would increase the latent heat flux into the boundary layer and have two major negative feedbacks on heat extremes: first, the increase in latent heat flux would be at the cost of the sensible heat flux, and a reduction in sensible heat flux would potentially reduce

any land amplification on heatwaves over forested regions. Second, by moistening the boundary layer, the chance of clouds being formed would increase, leading to a decrease in solar radiation at the surface and a consequent cooling effect.

In climate models, including CMIP5 models, the land surface is represented by modules that assume photosynthesis and $g_s$ (and consequently transpiration) are inherently coupled (De Kauwe et al., 2013). At high temperatures, models assume that

photosynthesis is reduced due to: (i) the direct impairment of the photosynthetic biochemistry; (ii) increased respiration; and (iii) reduced $g_s$ due to the associated high vapour pressure deficit. Finding additional evidence of a decoupling between photosynthesis and $g_s$ at high temperatures would therefore require revisiting existing assumptions embedded in all climate models and have important implications for model-based assessments of the role of the land surface in the amplification of heat extremes.

Here, we hypothesised that evidence of decoupling would present itself as a reduction in gross primary productivity (GPP) and an increase latent heat flux (LE) as air temperatures increased. It is important to clarify that decoupling does not mean that $g_s$ will increase as GPP declines, only that it will decline *less strongly* than current theory would predict if photosynthesis and $g_s$ remained coupled. As temperature increases, vapour pressure deficit (D) also increases, which will drive an increase in LE

unless there is stomatal closure, but this effect is unrelated to the decoupling mechanism we seek to find. To disentangle the potentially contributing role of D, we also explored these data based on the theoretical expectation (Lloyd et al. 1991; Medlyn et al. 2011; Zhou et al. 2014) that transpiration (E) is approximately proportional to GPP $\times$ $D^{0.5}$ (g C kPa$^{0.5}$ m$^{-2}$ d$^{-1}$; Eqn. 7). This expectation is based the idea of optimal stomatal behaviour proposed by Cowan and Farquhar (1977) that stomata should be regulated so as to maximise photosynthetic carbon gain for a given amount of transpiration. Medlyn et al. (2011) derived

the optimal stomatal behaviour as:

$$G_s = 1.6\left(1 + \frac{g_1}{\sqrt{D}}\right)\frac{A}{C_a} \tag{1}$$

where $G_s$ is canopy stomatal conductance to $CO_2$ (mol m$^{-2}$ s$^{-1}$), A is the net assimilation rate (μmol m$^{-2}$ s$^{-1}$), $C_a$ is the ambient atmospheric $CO_2$ concentration (μmol mol$^{-1}$), D is the vapour pressure deficit (kPa), the parameter $g_1$ (kPa$^{0.5}$) is a fitted parameter representing the sensitivity of the conductance to the assimilation rate and the factor 1.6 is the ratio of diffusivity of water to $CO_2$ in air. Assuming that transpiration is largely controlled by conductance, this relationship can be rearranged to

show that water-use efficiency (A/E) is approximately proportional to $1/\sqrt{D}$. This dependence has been remarked by many authors (e.g. Lloyd et al. 1991, Katul et al. 2009). Based on this dependence, Zhou et al. (2014, 2015) proposed an "underlying water-use efficiency" (uWUE) for eddy covariance data:

$$uWUE \approx \frac{GPP\sqrt{D}}{E}$$

(2)

Zhou et al. (2014) argued that the $D^{0.5}$ term provided a better linear relationship between GPP and E. Thus, to probe the effect of D, we focused on heatwaves (i.e. approach 2) and plotted LE expressed as evapotranspiration (mm day$^{-1}$), as a function of GPP×$D^{0.5}$. We note for the interested reader tracing the development of the optimal stomatal theory through the cited publications, that equation 7 in Medlyn et al. (2011) is missing a pressure (P) term in the numerator (under the square root sign), which ensures the equation is dimensionally correct. However, the equation is not used in any further derivation in Medlyn el. (2011) and so the missing term does not have any impact on the theory presented in the rest of that paper.

In this paper we therefore explore eddy-covariance measurements to examine whether there is widespread field-based evidence that during heat extremes, trees decouple photosynthesis and $g_s$, leading to increased transpiration. In contrast to previous experimental studies (e.g. Urban et al. 2017), our focus is on the ecosystem-scale and so we analysed the photosynthetic decoupling between photosynthesis and transpiration using theory derived from optimal stomatal behaviour (Lloyd et al. 1991; Medlyn et al. 2017; Zhou et al. 2014). We chose to focus on wooded ecosystems as the capacity to maintain transpiration throughout a heat extreme most likely requires deep soil water access and is in line with previous experimental evidence from trees (Drake et al., 2018; Urban et al., 2017).

## 2 Materials and Methods

### 2.1 Evidence of photosynthesis-transpiration decoupling

A number of experimental studies reporting photosynthetic decoupling have focused on the coupling between A and $g_s$ (Weston and Bauerle, 2007; Ameye et al. 2012; von Caemmerer and Evans, 2015), as opposed to A and E (Drake et al. 2018). At the ecosystem-scale (eddy-covariance), coincident measurements of $G_s$ and LE (or transpiration) are rarely available. Whilst it is possible to estimate the canopy $G_s$ by inverting the Penman-Monteith using measured LE, such an approach necessitates additional assumptions related to the canopy boundary layer conductance (Jarvis and McNaughton, 1986; De Kauwe et al. 2017), the canopy net radiation and the ground heat flux (Medlyn et al. 2017). Here we avoid these assumptions by focusing our analysis on the measured LE flux, as opposed to an estimate of the canopy $G_s$.

A range of definitions currently exist to identify an extreme temperature event (see Perkins et al. 2014 for a review). Most of these are defined from the context of the climate and may not reflect the physiological adaptations of the vegetation. Given this lack of a single unifying definition, we tested two approaches on the eddy-covariance measurements: (1) the change in GPP and latent heat flux during the four days leading up to and including a temperature extreme, where a temperature extreme was defined as being a day when the daily maximum temperature exceeded 37°C; and (2) the change in GPP and latent heat

flux during a heatwave, defined as at least three consecutive days where the maximum daily temperature exceeded 35°C. The first approach can be viewed as analogous to the behaviour leading up to the hottest day of the year (commonly defined as TXx; Klein et al. 2009) and the imposed lower boundary of 37°C similar to selecting a number of "hot" days by using a percentile from the TXx but defined from a more physiological standpoint. This temperature threshold was selected to ensure

the events were hot enough to stress the vegetation (Curtis et al. 2016; O'Sullivan et al. 2017; Zhu et al. 2018). For the Australian sites, 37°C was consistent with a site's climate-of-origin + threshold (mean summer maximum temperature; Tmax + 10°C) (Drake et al. 2017).

For each of these events we recorded the maximum daytime temperature, the mean daytime (6am – 8 pm) LE, and the daytime

summed GPP. Although we chose to compare mean daytime LE and the summed daytime GPP with the maximum daytime temperature, there are of course alternative analysis approaches. We chose our approach as an appropriate trade off in time resolution that facilitated us to consider several heat-extreme events, across multiple sites. This allowed us to see the broader patterns of behaviour at the ecosystem-scale. Had we considered analysing the raw 30-minute data for example, we felt that interpretation of the underlying behaviour would been made considerably more difficult due to the increased time frequency

and inherent noise in these data. A further alternative analysis approach would have been to compare the maximum or daily mean temperature with the midday GPP and LE fluxes; however, we felt such an approach could miss interesting morning and afternoon responses which may result directly from the temperature extremes but not be present in the midday observation.

To test for evidence of photosynthetic decoupling in the ecosystem-scale fluxes we fitted a linear regression to the fluxes from

each event leading up to a day where the maximum temperature exceeded 37°C (i.e. approach 1 above), showing events where the fitted slope was negative for GPP and positive for LE. We do not necessarily expect the response of GPP or LE to be linear with respect to increasing temperature, but selecting events based on their fitted (positive/negative) slopes allows us to identify patterns in the data. We do not seek to draw inference from the fitted slope being significant or not, given the small number of samples (n=4) in each event. We simply use this distinction to identify stronger positive or negative trends in these data. To

disentangle the potentially contributing role of D, we also explored flux behaviour based on the theoretical expectation (Lloyd et al. 1991; Medlyn et al. 2011; Zhou et al. 2014) that E) is approximately proportional to $GPP \times D^{0.5}$ (g C $kPa^{0.5}$ $m^{-2}$ $d^{-1}$). To address this issue, we focussed on heatwave events (i.e. approach 2 above).

## 2.2 Flux data

Half-hourly eddy covariance measurements of the exchange of carbon dioxide, energy, and water vapour were obtained from

the OzFlux (http://www.ozflux.org.au/) and FLUXNET2015 (http://fluxnet.fluxdata.org/data/fluxnet2015-dataset) and releases. We confined our FLUXNET2015 analysis to sites classified as wooded according to the International Geosphere–Biosphere Programme, namely: evergreen needleleaf forest; evergreen broadleaved forest; and deciduous broadleaved forest (albeit noting that these names have an inherently Northern Hemisphere bias, and would be better classified as evergreen

coniferous, evergreen angiosperm, and deciduous angiosperm forest, respectively). We excluded sites classified as savanna due to the associated complication of needing to attribute the total transpiration flux to grasses and trees; however, we do acknowledge that many of the Australian sites are also relatively open (see screening step below). We also excluded sites classified as mixed forest from our analysis, or those that did not meet our physiological threshold of a daily maximum temperature that exceeded 37°C. We also excluded sites that experienced burning. A total of nine sites met these criteria in the Tier 1 (freely available) FLUXNET 2015 database. FLUXNET data were pre-processed using the FluxnetLSM R package (Ukkola et al., 2017). For OzFlux, we used Level 6 gap-filled data following Isaac et al. (2017). These data were then screened to only keep measured and good-quality gap filled data. Events were ignored if a rainfall event greater than 0.5 mm day$^{-1}$ was observed during, or in the two days prior to a heat event in the eddy covariance data (Dekker et al., 2001; Law et al., 2002; Groenendijk et al., 2011; Keenan et al., 2013; Dekker et al. 2016; De Kauwe et al. 2017; Knauer et al. 2017; Medlyn et al. 2017) as this could bias the LE flux by leading to an increase in LE not associated with the mechanism we wished to identify (i.e. due to soil/canopy evaporation). Knauer et al. (2017) is the only study to have explicitly tested the impact of assuming that two days following a rainfall event, the LE flux can be assumed to dominated by transpiration. Across six FLUXNET sites, they found between a 9% and 19% change in estimates of the slope parameter of the optimal stomatal parameter ($g_1$; Medlyn et al. 2017) with increasing time since the last rainfall event beyond 48 hours (out to 240 hours). However, their analysis did not account for the potential confounding effect that as they screened a greater number of hours following rainfall, the number of samples used to estimate the $g_1$ parameter was also reduced, which would increase the error in estimates of the model parameter. Given both the high temperatures considered in our analysis framework and the length of the period after screening for rain (at least three days), we would expect the impact of soil evaporation to be a minor consideration.

## 2.3 Accumulated heat stress

To characterise a measure of the annual heat accumulated stress experienced by the vegetation we calculated the average number of growing degree days above our upper threshold of 37°C per year ($GDD_{37}$). We used surface air temperature from the 6-hourly, re-analysis by the Global Soil Wetness Project Phase 3 (GSWP3; http://hydro.iis.u-tokyo.ac.jp/GSWP3 and Dirmeyer et al. 2006) dataset during the period of 1970-2015 at a 0.5° spatial resolution. We opted to use this coarser dataset to estimate $GDD_{37}$ rather than the observed flux record due to the longer temporal record, which is likely to be more reflective of longer-term conditions.

## 2.4 Analysis code

All analysis code is freely available from: https://github.com/mdekauwe/heat_extremes_decoupling.

## 3 Results

We first focus on the Australian sites as these experienced more temperature extremes due to the warm climate. We found significant evidence of thermal heat stress (Table 1), with 85.8 $GDD_{37}$ at Alice Springs, 85.1 $GDD_{37}$ at Great Western Woodlands, 68.3 at Calperum, 31.7 at Gingin, 13.5 at Cumberland Plains, 13.4 at Whroo and 3.1 at Wombat.

Figure 1 shows a consistent reduction in the flux-derived GPP with increasing daily maximum temperature for each of the events (4-day events, where the maximum temperature > 37°C). We emphasise (see methods) that one should only interpret differences between significant negative slopes (dark blue lines) and negative slopes (dark green lines) as indicative of (possibly) stronger or more consistent reductions in GPP as a function of temperature. This reduction in GPP follows theory related to biochemical, respiratory and stomatal drivers (Lin et al., 2012). With the exception of the Whroo site, GPP was
reduced to close to zero at temperatures greater than 40°C. Figure S1 shows the limited occurrences where the fitted slopes indicated a positive (or arguably flat) response with increasing temperature.

Evidence for the hypothesised decoupling between photosynthesis and $g_s$, which would lead to an increase in LE with temperature (but a concomitant decline in GPP, Fig. 1), is shown in Fig. 2. Despite variability in the measured data, at each of the seven sites, LE is found to increase or be sustained as the temperature increases in the lead up to the maximum temperature
of each heat event. This increase is steepest at the Wombat State Forest site but is based on only one $GDD_{37}$ event (Table S1). At the other sites, the magnitude of the increase is smaller. However, it is clear that the LE flux is not reduced in line with GPP (Fig. 1) and instead remains sustained with temperature throughout the extreme events. Figure S2 shows the occurrences where the fitted slopes indicated a negative response with increasing temperature. In many cases these events were broadly flat in response to increasing temperature, again indicating a sustained LE flux. Taken together, Figs. 1, 2 and S2 provide consistent
evidence of a decoupling between photosynthesis and transpiration during significant heat extremes across a range of Australian wooded ecosystems.

We now seek to explore the strength of this apparent decoupling in more detail by looking at the ratio of positive to negative fitted slopes shown in Figs 1 and 2 and Figs S1 and S2. Figures 3 and 4 shows the distribution of fitted positive and negative slopes as a function of temperature across the Australian sites for GPP and LE, respectively. Whilst the fitted slopes for GPP
are predominately negative (Figure 3), there does not appear to be a consistent pattern in the frequency of positive vs. negative fitted LE slopes, with some sites having more positive slopes (e.g. Gingin, Great Western Woodlands) and some registering more negative slopes (Calperum, Whroo), while others are about even (Alice Springs, Cumberland Plains) (Figure 4). This result is not surprising given our hypothesis that significant transpiration during a heatwave is dependent upon the available supply of soil moisture. As soil water supply becomes limiting, we would expect to find more frequent negative slopes.
Consistent with this link to soil moisture, there is a small drop in the proportion of positive slopes (i.e. increased LE) towards the end of summer, which is coincident with an increase in the frequency of negative slopes (Fig S3).

Evidence for an increase in LE with temperature and for photosynthetic decoupling during heat extremes was much weaker across the seven FLUXNET2015 sites (excluding Australian sites; Fig. S4 and S5) that exceeded our 37°C threshold. The number of concomitant negative GPP slopes (Fig. S4) and positive LE slopes (Fig. S5) was noticeably lower when compared to Australian sites, making it harder to draw clear inferences. On the one hand, the weaker evidence from across the larger FLUXNET2015 dataset may point to this decoupling behaviour being species or climatic zone specific (i.e. located in very hot environments). However, we would caution against that interpretation as it is as likely to also point to the lack of representation of FLUXNET sites in regions, other than Australia, that experience very hot temperature extremes (e.g. the average $GDD_{37}$ for the non-Australian sites was >1 at only two sites, Table 1). Given the limited signal in the results obtained from FLUXNET2015 sites, we continue to focus our analysis on Australian sites. However, given the extremely hot summer experienced across Europe in 2018, future studies may wish to revisit this analysis as these updated flux data become available.

Increasing temperature also usually leads to increasing D and as a result, even with perfect coupling between photosynthesis and $g_s$, we would still expect to see transpiration changing as a function of $GPP \times D^{0.5}$. Figure 5 shows this relationship for consecutive heatwave and non-heatwave days (note Wombat State Forest was excluded from this analysis as there were insufficient consecutive days > 35°C.) If the change in transpiration was being driven by a decoupling of $g_s$ from the response of photosynthesis, we might expect to see increasing transpiration for a given $GPP \times D^{0.5}$, i.e. a spread in points vertically for heatwave days. If the change was being driven by increasing water use efficiency, we might expect to see an increased $GPP \times D^{0.5}$ for a given unit of transpiration, i.e. a spread horizontally for heatwave days. Across the sites there was not a clear difference in the behavior for heatwave vs. non-heatwave days. At Calperum, Cumberland Plains and Whroo the relationship between $GPP \times D^{0.5}$ and transpiration was fairly constant, whereas at Great Western Woodlands, transpiration for a given $GPP \times D^{0.5}$ on heatwave days was slightly higher than on non-heatwave days and at Alice Springs and Gingin, slightly lower. At Alice Springs and Gingin, this seems to fit with our expectation of increasing D driving increasing water use efficiency, i.e. not the decoupling mechanism. At Great Western Woodlands, there is some indication the data spread vertically, which may be consistent with our expectation outlined for decoupling, but the pattern is not conclusive.

**4 Discussion**

Recent experimental studies (Drake et al., 2018; Urban et al., 2017) have identified that at very high temperatures (> 40°C), plant decouple photosynthesis and $g_s$ and instead increase transpiration in an apparent active process to cool their canopies. Our results from across seven wooded ecosystems located in Australia were inconclusive. We found some indication (Figs. 1-4) that LE was increased or sustained as GPP decreased when exploring the behavior in the lead up to the hottest days of the year. However, when we focused on heatwave events (i.e. consecutive days > 35°C; Fig. 5) and considered the role of D, i.e.

as a driver of increased LE, rather than a photosynthetic-decoupling that would increase the transpiration flux to cool the canopy (i.e. in response to leaf temperature), we found little clear support for photosynthetic decoupling.

As the background climate warms with associated changes in the intensity and frequency of heat extremes, there is a growing interest in the degree to which leaf temperature affects, and is affected by, the physiological response of plants. The potential for plants to use a photosynthetic decoupling mechanism to apparently regulate leaf temperatures is one emerging aspect of this interplay between plant physiology and temperature. Other studies are currently questioning other widely-accepted notions about stomatal regulation. For example, Cernusak et al. (2018) recently examined the near universal assumption that vapour pressure inside a leaf remains saturated in all conditions. They found in two conifer species that, under moderate to high D, this assumption was invalid leading to a bias in the calculated $g_s$. Similarly, Kowalski et al. (2017) have recently questioned the paradigm that all transport through stomata is diffusive, instead invoking the concept of non-diffusive stomatal jets. However, neither of these papers provides a mechanism by which stomatal closure would be decoupled from photosynthesis. Further plant physiological studies are required to identify this mechanism.

### 4.1 Why did we not find supporting evidence for ecosystem-scale photosynthetic decoupling?

One interpretation of the apparent contradictions between the findings of previous studies and our lack of conclusive evidence at the ecosystem-scale, may simply relate to the interpretation scale. At the leaf-level, plants usually reduce $g_s$ exponentially with increasing D (Oren et al. 1999). However, at high temperatures and with the associated high D, the increased atmospheric demand for water may drive an increase in the transpiration rate. In well-controlled environments, it may be possible to separate the direct response to temperature from that of D, but as our analysis shows, this is more complicated with ecosystem-scale data.

The recent work by Drake et al. (2018) demonstrated clear evidence of photosynthetic decoupling at the canopy scale using a series of whole-tree chambers, which would suggest that this mechanism is unlikely to simply be scale dependent. However, to infer the photosynthetic decoupling, Drake et al. (2018) demonstrating that the observed decline in $g_s$ (and so transpiration) was weaker than predicted by a coupled leaf A-$g_s$ model, which was specifically calibrated to the experimental data. This approach is not viable across multiple sites as it necessitates detailed site measurements for calibrations that are often prohibited by the tall canopy height of mature stands. Applying such a coupled model (e.g. a land surface model) to these site data simply demonstrates that the model is unable to capture the observed site responses (not shown). As a result, we could not reliably infer that the divergence from model behaviour points to evidence of photosynethetic-decoupling, as opposed to, for example, poor parameterization associated with stand level attributes such as leaf area index or root zone soil moisture.

A number of the previous studies that showed photosynthetic decoupling experimentally were carried out on well-watered plants (Ameye et al. 2012; Urban et al. 2017). Thus, one interpretation of our results is simply that root-zone soil moisture was

limiting any photosynthetic decoupling. In Drake et al. (2017), irrigation of the whole-tree chambers was withheld for the month prior to the heatwave experiment, thus a more nuanced interpretation may be that a photosynthetic decoupling mechanism requires access to soil moisture from deeper in the profile (perhaps associated with access to groundwater). Without data throughout the root-zone profile across the flux sites we cannot rule out this explanation. Our results did show tentative evidence consistent with this explanation; we found a small decrease in the number of positive slopes (i.e. increased LE) towards the end of the summer (Fig S3), which may reflect reduced soil water availability to sustain transpiration. Using sap flow data, Tatarinov et al. (2015) found a ~60% decrease in canopy conductance, an approximately halving of daytime GPP, but little change in ET during spring heat waves ('hamsin') in a 50-year-old Alepp pine forest located at the edge of the Negev desert. The observed responses during these Mediterranean heat extremes are consistent with a photosynthetic decoupling although in their study, we note that the authors attributed these differences in behavior to the relative influence of D and soil moisture availability.

One could ask whether our analysis considered hot enough temperatures (> 37°C) to trigger a photosynthetic decoupling mechanism. For example, during an imposed heatwave, Ameye et al. (2012) probed the decoupling mechanism at daily maximum temperatures between 47 and 53°C. Similarly, Zhu et al. (2018) found that most of the 62 species sampled across Australia exhibiting maximum critical temperatures near 50°C. However, the temperature optima for leaf and canopy photosynthesis in Eucalypts in southern Australia are well below 30 degrees (see Duursma et al. 2014; Drake et al. 2016; Kumarathunge et al. 2019), suggesting that days above 37°C should induce a decline in GPP. Our analysis also included events with daily maximum temperatures of greater than 40°C and consecutive heatwave days > 35°C. Therefore, we would argue that insufficiently high temperatures are unlikely to explain why we did not see clear evidence when looking at eddy covariance data.

Our approach relies on GPP which is not directly observed but is instead modelled using assumptions related to the extrapolation of night-time respiration and measured net ecosystem exchange. It is debatable whether these assumptions hold at very high temperatures, and examining these modelled GPP estimate estimates at high temperatures warrants further investigation, particular as researchers leverage these data to explore the responses of the vegetation to temperature extremes. Eddy-covariance data are also known to have issues closing the energy balance (Wilson et al. 2002; Foken 2008; Hendricks-Franssen et al. 2010; Eder et al. 2015), which may introduce errors into the LE flux (see Wohlfahrt et al. 2009, for a detailed discussion). For the seven Australian flux sites that make up the majority of our analysis, we calculated the ratio of the sum of latent and sensible heat fluxes to the sum of the net radiation and ground heat flux, finding on average a ~17% imbalance in the ratio (range 7-30%). Importantly however, we did not find any difference in this imbalance in heatwave vs. non-heatwave days. This is in line with other studies   Despite these limitations, FLUXNET eddy covariance flux measurements still present our best ecosystem-scale estimates of vegetation responses to heat extremes and have been widely analysed to address these types of questions (Ciais et al. 2005; Teuling et al. 2010; Wolf et al. 2013; von Buttlar et al. 2018; Flach et al. 2018).

Our analysis is also limited by the number of extreme events recorded in the existing record and the clear bias in these data towards Australian sites is due to the lack of representation of sites within the FLUXNET data collection that sample locations in extreme environments outside of Australia. In our analysis we focused on hot days and heatwaves with a very hot temperature range, i.e. consecutive days > 35°C, hence a fair criticism of our approach is that a lower threshold might be also

relevant for different environments and species. Any choice of threshold is arguably arbitrary; we chose ours to ensure we were focusing on the vegetation response to a threshold that would lead to a degree of physiological limitation and is in line with studies that suggest this occur at temperatures above our chosen thresholds (Curtis et al. 2016; O'sullivan et al. 2017; Zhu et al. 2018).

Although Drake et al. (2018) did not find evidence of increased litterfall during their heatwave experiment, it is of course

possible that across the FLUXNET sites we considered, there was some reduction in leaf area in response to high temperature extremes. However, any leaf area reduction would reduce both transpiration and photosynthesis and thus, we think it is unlikely to affect ecosystem-scale estimates of a photosynthetic decoupling. Nevertheless, future flux-based experiments may consider also using leaf litter traps at sites to allow researchers to separate out this effect and confirm this assumption.

Finally, throughout our manuscript we have treated the measured LE flux as interchangeable for the transpiration flux (i.e.

ignoring any potential role of soil and or canopy evaporation – see Methods 2.2). Strictly, if soil and/or canopy evaporation fluxes were not zero, the signal that we have analysed would contain a contribution that is not directly under the plants control and so could not be affected (directly) by any photosynthetic decoupling. Whilst we cannot rule out such a contribution we expect it to be unlikely to be a significant factor at play in our results. Screening the eddy covariance timeseries for the two days following observed rain events follows a widely used strategy in eddy covariance studies (Dekker et al., 2001; Law et al.,

2002; Groenendijk et al., 2011; Keenan et al., 2013; Dekker et al. 2016; De Kauwe et al. 2017; Knauer et al. 2017; Medlyn et al. 2017). Moreover, our analysis also considers heat extremes that last for at least three further days. Thus, after five days (two days prior to event must also have been rain free), in temperatures exceeding 30°C, we think it likely that the latent heat flux will be dominated by transpiration.

## 4.2 Implications for models

The potential implications for modelling studies that focus on heat extremes are clear, particularly for studies in Australia. None of the current generation of land surface models have the capacity to decouple transpiration from the down-regulation of photosynthesis with increasing temperature. Instead models assume photosynthesis and $g_s$ (and consequently transpiration) remain coupled at all times. As a result, climate models may underestimate the capacity of the vegetation to dampen heat extremes in simulations for Australia. This is also true of more sophisticated plant hydraulic models (Williams

et al. 2001) and profit-maximisation approaches (Wolf et al. 2016; Sperry et al. 2016) that hypothesise the cost of water is not fixed in time, but instead increases with increasing water stress. For these latter approaches to account for a

photosynthetic decoupling they would need to prioritise maintaining an optimum canopy temperature above a net carbon gain. However, mechanisms to capture this within models should likely wait for further supporting evidence of photosynthetic decoupling.

## 5. Conclusion

A number of recent experimental studies have highlighted that during heat extremes, plants may decouple photosynthesis and transpiration: reducing photosynthesis to near zero but increasing transpiration into the boundary layer. In this study we used eddy-covariance measurements to examine the evidence for a photosynthetic-decoupling in wooded ecosystems at the ecosystem-scale during heat extremes. When focussing on the three days leading up to a temperature extreme (a daily maximum > 37°C), we found some evidence of reduced photosynthesis and sustained or increased latent heat fluxes in seven

Australian evergreen wooded flux sites. However, when considering the role of vapour pressure deficit, we were unable to conclusively disentangle photosynthetic-decoupling from the effect of increase in transpiration due to increasing vapour pressure deficit during heatwaves (three or more consecutive days above 35°C). However, it would be premature to interpret our results as evidence that such a mechanism does not scale from the leaf to ecosystem. Instead, understanding the response of transpiration during heatwaves remains an important issue to resolve. It is clear that further experimental results will be

required to resolve this issue and these studies will need to be able to more clearly separate the decoupling mechanism from the response to D and other potential factors (see 4.1). To make progress on this photosynthetic-decoupling issue will likely require concurrent leaf-level gas-exchange measurements (photosynthesis and $g_s$) as well as canopy/ecosystem-scale transpiration. To date, most of our insight has been limited to the leaf-scale (Ameye et al. 2012; von Caemmerer and Evans, 2015; Urban et al. 2017), or a single canopy-scale study situated in whole-tree chambers (Drake et al. 2018). To bridge this

gap in our knowledge, it would be desirable to align future experiments with measurements taken at eddy covariance sites (i.e. by using matching species) to allow us to more easily test whether this mechanism scales to the ecosystem.

*Author contributions.* MDK conceived and designed the study based on discussions involving MDK, BEM and JED. MDK

wrote the code and analysed the results. AU assembled and processed the eddy covariance data. All authors contributed to writing of the paper.

*Code availability.* All code is freely available from: https://github.com/mdekauwe/heat_extremes_decoupling

*Data availability.* All Eddy covariance data are available from: http://www.ozflux.org.au/ and http://fluxnet. fluxdata.org/data/fluxnet2015-dataset

*Competing interests*. The authors declare no competing financial interests.

*Acknowledgements*. MDK acknowledges support from the Australian Research Council Centre of Excellence for Climate Extremes (CE170100023). This work used eddy covariance data acquired by the FLUXNET community and in particular by the following networks: AmeriFlux (U.S. Department of Energy, Biological and Environmental Research, Terrestrial Carbon Program (DE–FG02–04ER63917 and DE–FG02– 04ER63911)), AfriFlux, AsiaFlux, CarboAfrica, CarboEuropeIP, CarboItaly, CarboMont, ChinaFlux, Fluxnet–Canada (supported by CFCAS, NSERC, BIOCAP, Environment Canada, and NRCan), GreenGrass, KoFlux, LBA, NECC, OzFlux, TCOS–Siberia, USCCC. We acknowledge the financial support to the eddy covariance data harmonization provided by CarboEuropeIP, FAO–GTOS–TCO, iLEAPS, Max Planck Institute for Biogeochemistry, National Science Foundation, University of Tuscia, Université Laval and Environment Canada and US Department of Energy and the database development and technical support from Berkeley Water Center, Lawrence Berkeley National Laboratory, Microsoft Research eScience, Oak Ridge National Laboratory, University of California, University of Virginia.

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

30

**Figures**

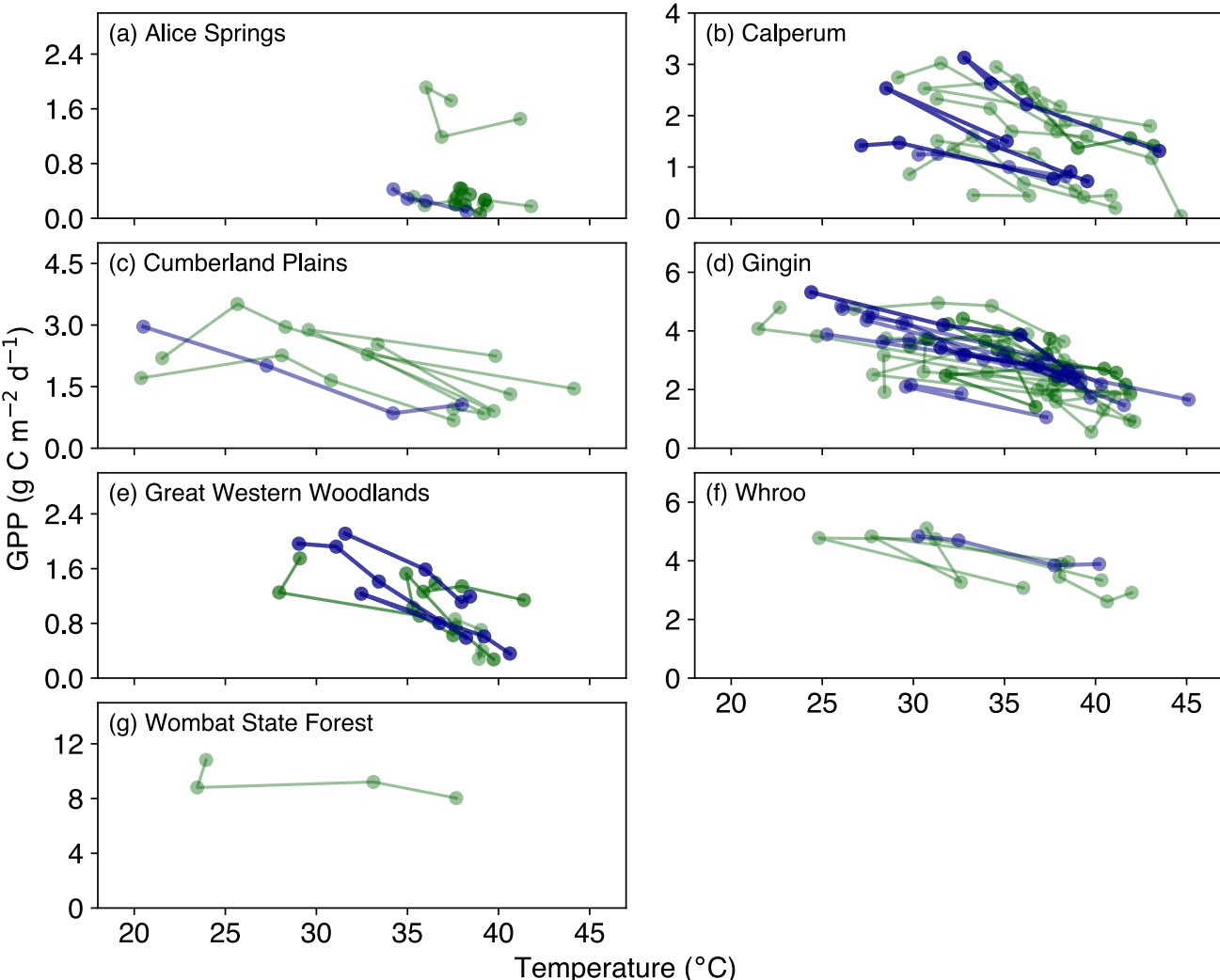

Figure 1: Evolution of GPP in the three days prior to and including a hot temperature extreme (daily maximum temperature exceeded 37°C). Dark blue lines represent events in which a fitted linear regression indicated a significant negative slope, whilst dark green lines represent events where the fitted slope was negative but not significant. Note in both cases, we are not showing the fitted slopes, we are simply using this approach to identify stronger positive or negative trends in these data (see methods). Events where the fitted slope was positive are shown in Figure S1.

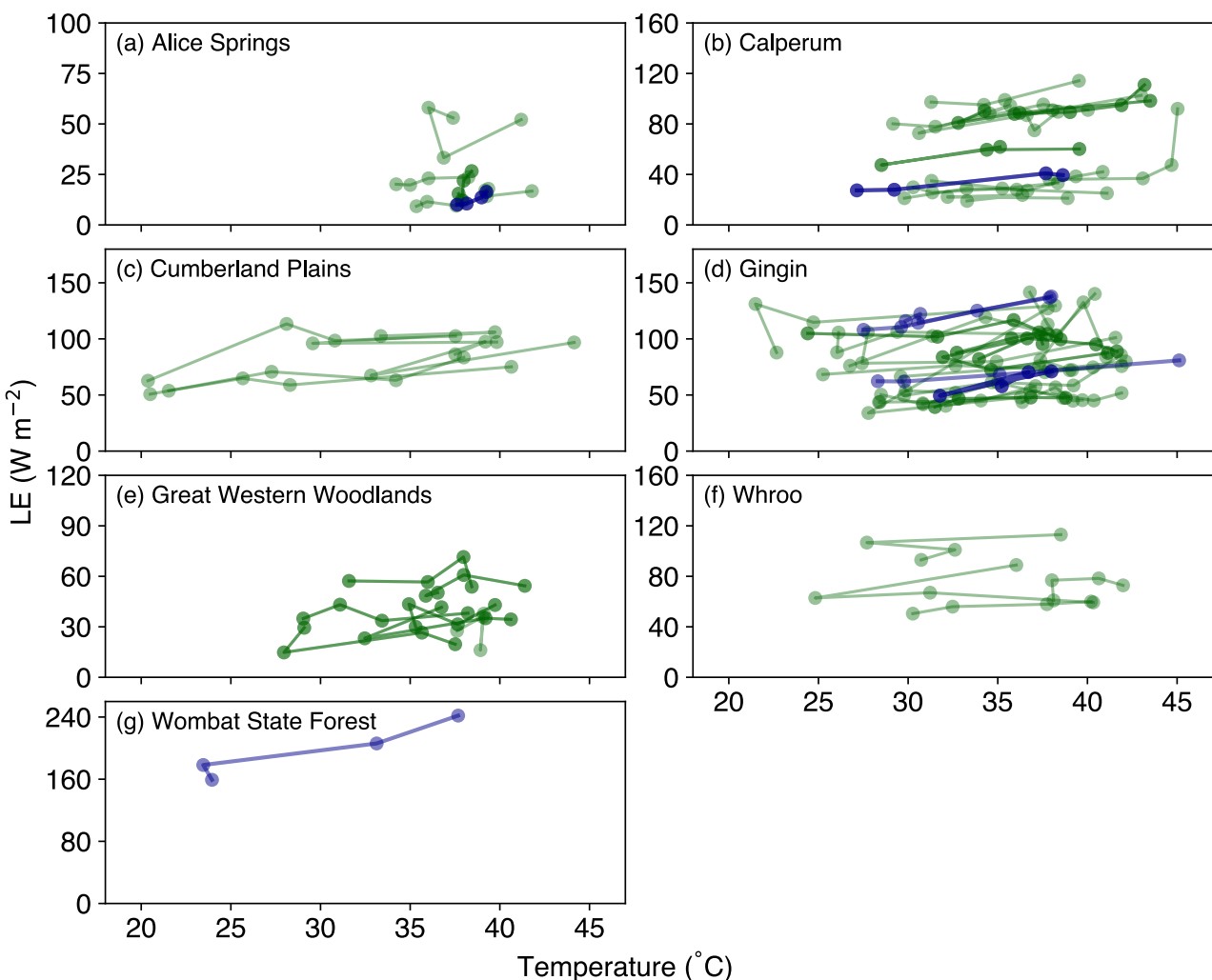

Figure 2: Evolution of LE in the three days prior to and including a hot temperature extreme (daily maximum temperature exceeded 37°C). Dark blue lines represent events in which a fitted linear regression indicated a significant positive slope, whilst dark green lines represent events where the fitted slope was positive but not significant. Note in both cases, we are not

5     showing the fitted slopes, we are simply using this approach to identify stronger positive or negative trends in these data (see methods). Events where the fitted slope was negative are shown in Figure S2.

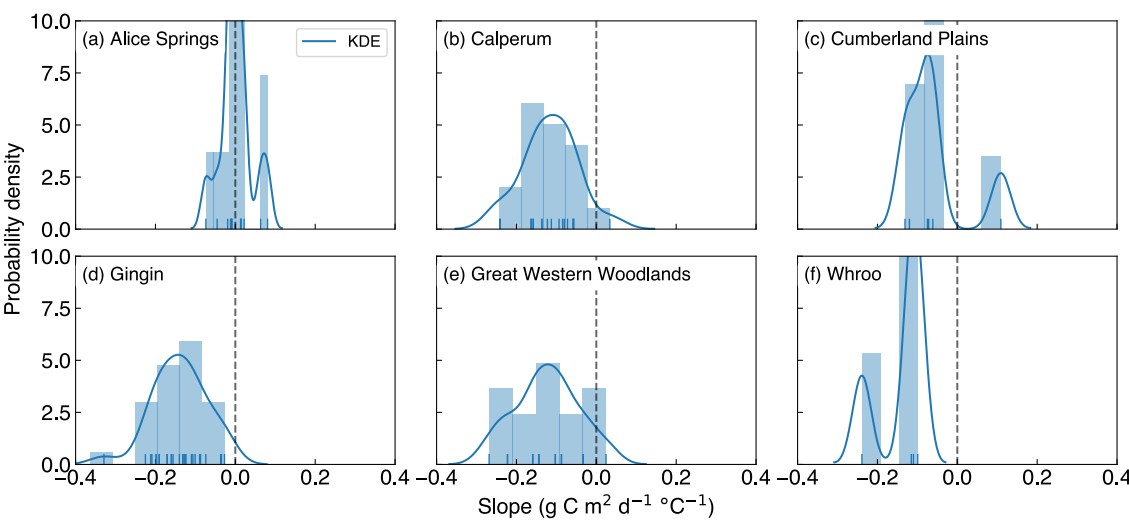

Figure 3: Probability density and histogram showing the distribution of fitted positive and negative GPP slopes across the OzFlux sites. The dark blue curve shows the fitted kernel density estimate (KDE) and the vertical blue lines along the x-axis are "rugs", which represent the individual occurrence of fitted slopes. Data from Wombat State Forest has been omitted from the graph as there was only one slope. Note, the sum of the bars can exceed one as the normalisation ensures that the sum of the bar heights multiplied by the bar widths equals one, which allows the normalised histogram to be compared to the KDE, which is normalised so that the area under the curve equals 1.

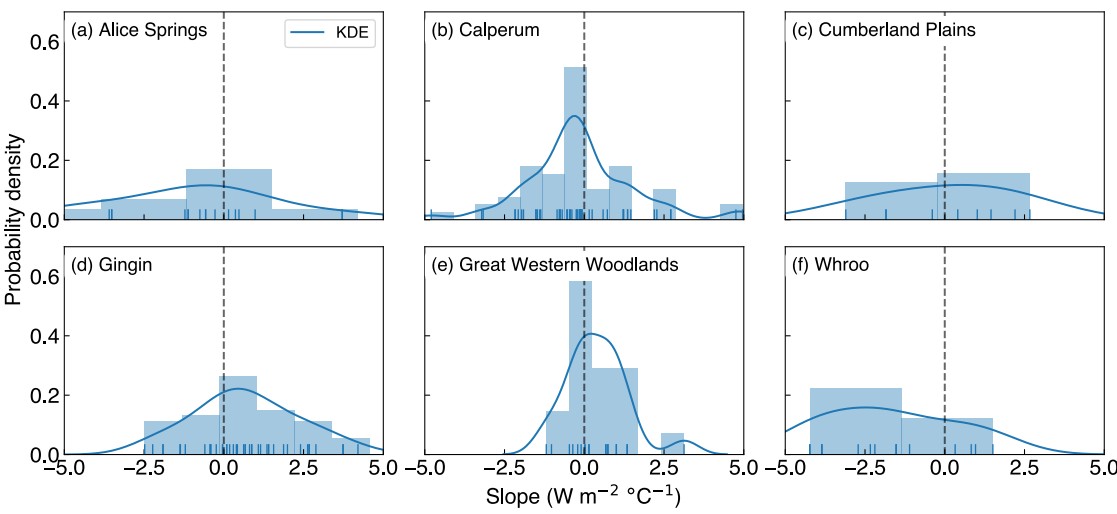

Figure 4: Probability density and histogram showing the distribution of fitted positive and negative LE slopes across the OzFlux sites. The dark blue curve shows the fitted kernel density estimate (KDE) and the vertical blue lines along the x-axis are "rugs", which represent the individual occurrence of fitted slopes. Data from Wombat State Forest has been omitted from the graph as there was only one slope. Note, the sum of the bars can exceed one as the normalisation ensures that the sum of the bar heights multiplied by the bar widths equals one, which allows the normalised histogram to be compared to the KDE, which is normalised so that the area under the curve equals 1

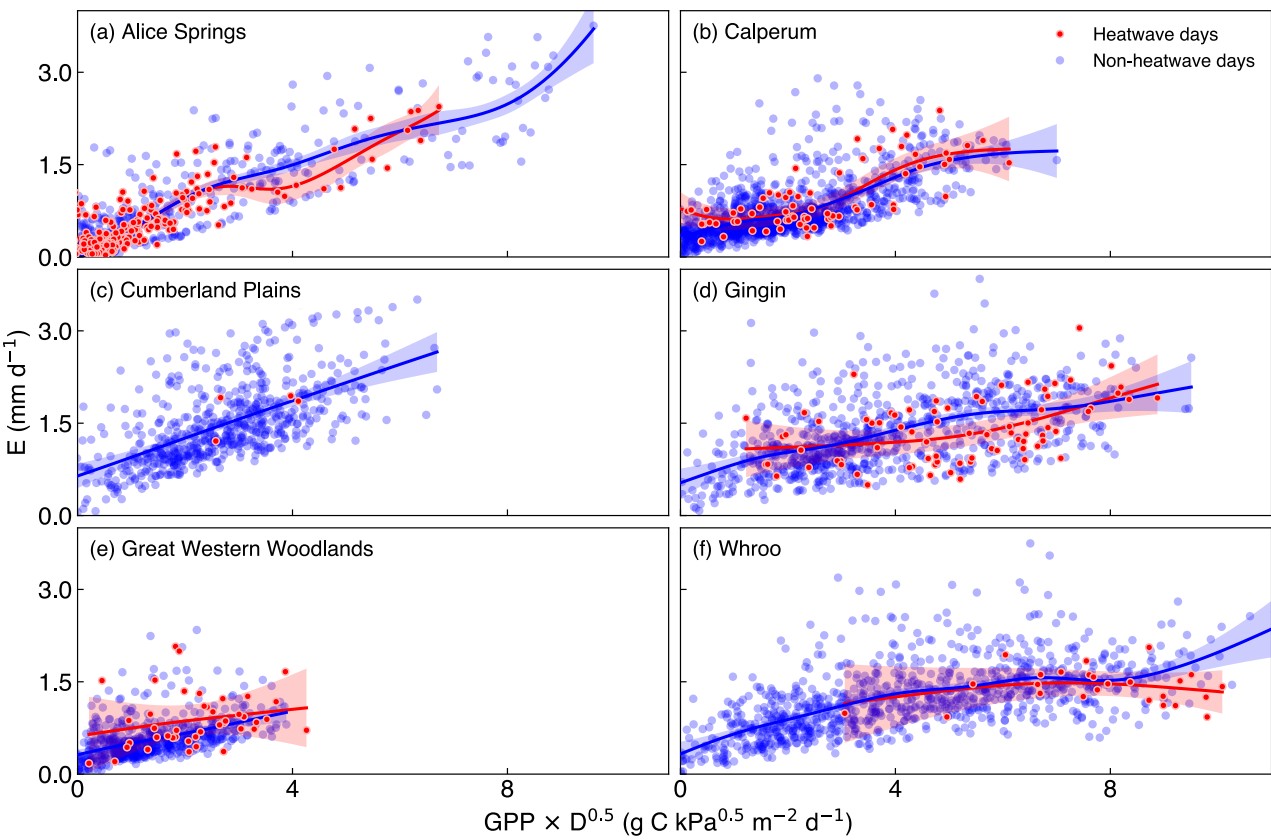

Figure 5: E as a function of GPP x $D^{0.5}$ on heatwave and non-heatwave days. The solid lines are smoothed time series using a generalized additive model (with a 95% confidence intervals). Note, data from Wombat State Forest has also been omitted from the graph as there were insufficient heat wave events and the generalized additive model was not fit to the heatwave days at Cumberland Plains due to the limited data.

**Tables**

Table 1. OzFlux and FLUXNET2015 site information. GDD$_{37}$ is the average number of growing degree days above our threshold of 37°C. Plant Functional types (PFT) were: ENF - evergreen needleleaf forest; EBF - evergreen broadleaved forest; and DBF - deciduous broadleaved forest. Note the FLUXNET sites Castel d'Asso 1 and 3 and Roccarespampani 1 and 2 have been combined in the table.

| Site | Latitude | Longitude | GDD$_{37}$ | Dominant Species | Years | Reference |
|---|---|---|---|---|---|---|
| ***OzFlux sites*** | | | | | | |
| Alice Springs | -22.28 | 133.25 | 85.8 | *Acacia aneura* | 2010-2013 | Cleverly et al. (2013) |
| Calperum | -34.00 | 140.59 | 68.3 | Mallee | 2011-14 | Meyer et al. (2015) |
| Cumberland Plains | -33.61 | 150.72 | 13.5 | *Eucalyptus moluccana and Eucalyptus fibrosa* | 2013-14 | Renchon et al. (2018) |
| Gingin | -31.37 | 115.71 | 31.7 | *Banksia* | 2012-14 | Silberstein (2015) |
| Great Western Woodlands | -30.19 | 120.65 | 85.1 | *Eucalyptus salmonophloia* | 2013-15 | Macfarlane (2013) |
| Whroo | -36.67 | 145.03 | 13.4 | *Eucalyptus microcarpa and Eucalyptus leucoxylon* | 2012-14 | McHugh et al. (2017) |
| Wombat State Forest | -37.42 | 144.09 | 3.1 | *Eucalyptus obliqua, Eucalyptus radiata and Eucalyptus rubida* | 2011-14 | Griebel et al. (2016) |
| ***FLUXNET2015 sites*** | | | | | | |
| Castel d'Asso | 42.38 | 12.02 | 0.1 | Poplar species | 2011-14 | Sabbatini et al. (2016) |
| Le Bray | 44.71 | -0.77 | 0.5 | *Pinus pinaster* | 1996-08 | Berbigier et al. (2001) |
| Mongu | -15.44 | 23.25 | 52.6 | *Brachystegia bakeriana and Brachystegia spiciformis* | 2000-09 | Merbold et al. (2009) |
| Morgan Monroe State Forest | 39.32 | -86.41 | 21.5 | *Acer saccharum, Liriodendron tulipifera, Sassafras albidum, Quercus alba and Quercus nigra* | 1999-2014 | Schmid et al. (2000) |
| Puechabon | 43.74 | 3.60 | 0.4 | *Quercus ilex* | 2000-14 | Rambal et al. (2004) |
| Qianyanzhou | 26.74 | 115.05 | 20.9 | *Pinus elliottii and Pinus massoniana* | 2003-05 | Yu et al. (2006) |

| Roccarespampani | 42.39 | 11.92 | 0.2 | *Quercus Cerris L.* | 2000-12 | Rey et al. (2002) |