# Peer review of "Examining the evidence for decoupling between photosynthesis and transpiration during heat extremes"

_Biogeosciences, 2018_

## Referee Comment (RC1) · A. Kowalski (Referee) · 5 Oct 2018

Prompted by recent observations from chamber measurements of a decoupling between photosynthesis and transpiration at high temperatures, De Kauwe and colleagues examine eddy covariance flux data to see whether such decoupling can be observed at the ecosystem scale. To my mind, this manuscript suffers from several important inadequacies, and requires major revision before it would be acceptable for publication. Anticipating that some of my criticisms will be viewed as controversial, I will nonetheless lay them all out, so that the editor can determine which (if any) deserve to be taken into consideration:

1. Both Tier-1 FLUXNET2015 data and OzFlux data suffer doubts regarding their

validity due to their persistent failure to demonstrate conformity with the principle of energy conservation (i.e., to close the surface energy budget). Although it might be going too far to say that it is inappropriate to download and analyze such data as the authors have done, neither do I think it is correct for this issue to be neglected entirely. Specifically, I am not aware that anyone has looked at the effect of heat waves on the energy balance closure, but this would certainly seem to be germane to the scientific questions that the authors are posing in the context of dataset validity. Also, although the FLUXNET2015 database includes a GPP variable, this is not measured by flux towers and the procedure from which it is inferred is of dubious validity during conditions of extreme heat stress. Given that the authors are attempting to tease out subtle temperature dependencies of GPP (which is not measured directly) and LE (which fails energy conservation checks), it seems inappropriate to me that such issues are not mentioned at all in this paper.

2. The paper draws no concrete conclusions, partly I think because the organisation of the manuscript is below standard. The paper contains about 1 page of introduction, 1.5 pages of methods, and 2.5 pages of "Results and discussion" to which will be added five figures and a table. This last section makes for difficult reading, in part because the authors appear to make little effort to distinguish between the facts and their interpretations thereof. Furthermore, the paper contains no equations whatsoever, despite the fact that the authors plot a variable (the product of GPP and the square root of the vapour pressure deficit) whose grouping cannot be justified (see comment number 3 below). All of these structural shortcomings make it particularly difficult for the reader to extract and evaluate the underlying message of the manuscript. I believe that the paper would be much better organised with a classical structure of 1. Introduction 2. Methods 3. Results 4. Discussion & 5. Conclusions.

3. According to the abstract, an important aspect of the paper addresses "the role of vapour pressure deficit" (D). The authors describe this in terms of the "theoretical expectation of the effect of D on g_s" (page 3, line 27), citing previous works in this

regard. Although not explicitly appearing in this manuscript, the "equation" underlying this idea is eq. (7) from the 2011 paper by Medlyn et al., which is demonstrably incorrect. One of the major contributions to science of Joseph Fourier is the criterion of "dimensional homogeneity", which states that only quantities with the same dimension can be compared, equated, added or subtracted. An obvious example would be the ridiculous statement that one kilometer is greater than one second. At the risk of sounding harsh, I must point out that equation (7) of the Medlyn et al. (2011) paper is equally absurd, and should not be considered as a "theoretical expectation". This absurdity seems to me to be a likely explanation for the fact that no units are included on the abscissa of Figure 5 of the De Kauwe et al manuscript, defined by a combination of variables (again: the product of GPP and the square root of the vapour pressure deficit; since it would be fitting for such a group of variables to be defined and assigned a symbol, I will call it Beta). The units of Beta would necessarily include the square root of a pressure unit such as mb or Pa (equivalent to the square root of a $kg\ m^{-1}\ s^{-2}$). My guess is that the unpleasantness of such a unit caused it to be excluded in the axis label. I would argue that Beta should be rejected altogether based on the powerful tool of dimensional analysis, which invalidates eq. (7) from the 2011 Medlyn et al. paper.

4. The ordinates of figures 3 and 4 are labelled with "density", a variable that normally would have units such as $kg\ m^{-3}$. Rather, I believe that what the authors have plotted is a frequency of occurrence, which is a fractional, non-dimensional quantity that requires no units. However, since the values in figure 3 go well above unity, I suspect that they should be described in terms of percent (%). In any event, I think this needs to be clarified.

5. (This final comment may be viewed by the editor as excessively ego-centric on the part of the reviewer. Nonetheless I feel obligated to point it out.) I have applied the laws of physics to demonstrate that the paradigm underlying the definition of the "stomatal conductance" is fundamentally incorrect (Kowalski, Atmos. Chem. Phys., 17, 8177–8187, 2017), and furthermore to *predict* a decoupling of transpiration and

photosynthesis at high temperatures. The long-standing paradigm in ecophysiology presupposes all transport through stomata to be diffusive in nature, whereas my analysis, based on conservation of linear momentum, shows that non-diffusive transport also occurs in the form of "stomatal jets". In brief, because the exchange of water vapour dominates surface exchange of all gases, the evaporation rate defines a flow velocity away from the evaporating surface and consequent transport of all gases away from the evaporating surface. For the particular case of water vapour, the analysis shows that the specific humidity represents the fraction of water vapour transport that is non-diffusive. Students of thermodynamics know that, for a saturated environment such as that supposed by ecophysiologists within a stomatal cavity, the specific humidity increases nearly exponentially as a function of temperature. Thus, at extreme temperatures the role of non-diffusive transport becomes non-negligible and a decoupling is expected between exchanges of water vapour (whose egress is aided by non-diffusive transport) and carbon dioxide (whose ingress is opposed by the outgoing Stefan flow). At the extreme limit of the boiling point, the vapour pressure inside the stomatal cavity would equal the total air pressure, meaning that (1) water vapour would be the lone gas inside the stomatal cavity, therefore (2) no diffusion could occur, and all transport would be non-diffusive (i.e., a specific humidity of 100%), and therefore (3) no photosynthesis would be possible (with no $CO_2$ present). Since my analysis is soundly based on the laws of physics and satisfactorily explains the decoupling between photosynthesis and transpiration at high temperatures, I believe that the authors should take it into account when exploring this "previously overlooked vegetation-atmosphere feedback that may in fact dampen, rather than amplify, heat extremes". However, I hardly think it is my place to insist that other scientists cite my papers, and so must leave judgement of this matter to the editor.
* * *

---

## Author Comment (AC1) · 10 Oct 2018

We thank the reviewer for their constructive comments and we address their various concerns below. Referee comments are highlighted in bold, with our response below in each case.

**Prompted by recent observations from chamber measurements of a decoupling be- tween photosynthesis and transpiration at high temperatures, De Kauwe and col- leagues examine eddy covariance flux data to see whether such decoupling can be observed at the ecosystem scale. To my mind, this manuscript suffers from several important inadequacies, and requires major revision before it would be acceptable for publication. Anticipating that some of my criticisms will be viewed as controversial, I will nonetheless lay them all out, so that the editor can determine which (if any) deserve to be taken into consideration:**

**1. Both Tier-1 FLUXNET2015 data and OzFlux data suffer doubts regarding their validity due to their persistent failure to demonstrate conformity with the principle of energy conservation (i.e., to close the surface energy budget). Although it might be going too far to say that it is inappropriate to download and analyze such data as the authors have done, neither do I think it is correct for this issue to be neglected entirely. Specifically, I am not aware that anyone has looked at the effect of heat waves on the energy balance closure, but this would certainly seem to be germane to the scientific questions that the authors are posing in the context of dataset validity. Also, although the FLUXNET2015 database includes a GPP variable, this is not measured by flux towers and the procedure from which it is inferred is of dubious validity during conditions of extreme heat stress. Given that the authors are attempting to tease out subtle temperature dependencies of GPP (which is not measured directly) and LE (which fails energy conservation checks), it seems inappropriate to me that such issues are not mentioned at all in this paper.**

We appreciate the Reviewers concerns on this issue.

We note in response to their statement about GPP that on page 6 of our submission that we stated: "*Our analysis also relies on GPP which is not directly observed but is instead modelled using assumptions related to the extrapolation of night-time*

*respiration (ER) and measured net ecosystem exchange. It is debatable whether these*
*assumptions hold at very high temperatures, and examining these modelled GPP*
*estimate estimates at high temperatures warrants further investigation particular as*
*researchers leverage these data to explore the responses of the vegetation to*
*temperature extremes."*

In revision, we will add a caveats section to our new discussion section (see next response) where we will discuss issues related to the GPP data and the energy balance closure issue in relation to the latent heat flux. Furthermore, despite caveats, eddy covariance data represent one of our key constraints on the carbon, energy and water cycles and are regularly used to probe ecosystem responses to extremes (e.g. von Buttlar, et al. 2018: Impacts of droughts and extreme-temperature events on gross primary production and ecosystem respiration: a systematic assessment across ecosystems and climate zones, Biogeosciences, 15, 1293-1318).

**2. The paper draws no concrete conclusions, partly I think because the organisation of the manuscript is below standard.**

We would disagree with this interpretation. In our paper we tested whether a photosynthetic decoupling mechanism identified in whole-tree chamber experiments (e.g. Drake et al. 2018, Global Change Biology) was present at the ecosystem scale. As our results demonstrate, outside of the experimental environmental, it is difficult to isolate such a mechanism. In so far as we can draw conclusions from the FLUXNET data, we did not find strong support for the original experimental result. However, absence of evidence is not evidence of absence and as result, to be more concrete with our conclusions given some the caveats of the data felt unwarranted. As a result, we discussed the need for new field-based studies to tackle this issue further.

**The paper contains about 1 page of introduction, 1.5 pages of methods, and 2.5 pages of "Results and discussion" to which will be added five figures and a table. This last section makes for difficult reading, in part because the authors appear to make little effort to distinguish between the facts and their inter- pretations thereof. Furthermore, the paper contains no equations whatsoever, despite the**

**fact that the authors plot a variable (the product of GPP and the square root of the vapour pressure deficit) whose grouping cannot be justified (see comment number 3 below). All of these structural shortcomings make it particularly difficult for the reader to extract and evaluate the underlying message of the manuscript. I believe that the paper would be much better organised with a classical structure of 1. Introduction 2. Methods 3. Results 4. Discussion & 5. Conclusions.**

We are happy to reorganise our manuscript as suggested by the reviewer and this will allow us to tackle the issue they highlighted in their first comment.

**3. According to the abstract, an important aspect of the paper addresses "the role of vapour pressure deficit" (D). The authors describe this in terms of the "theoretical expectation of the effect of D on g_s" (page 3, line 27), citing previous works in this regard. Although not explicitly appearing in this manuscript, the "equation" underlying this idea is eq. (7) from the 2011 paper by Medlyn et al., which is demonstrably in- correct. One of the major contributions to science of Joseph Fourier is the criterion of "dimensional homogeneity", which states that only quantities with the same dimen- sion can be compared, equated, added or subtracted. An obvious example would be the ridiculous statement that one kilometer is greater than one second. At the risk of sounding harsh, I must point out that equation (7) of the Medlyn et al. (2011) paper is equally absurd, and should not be considered as a "theoretical expectation". This ab- surdity seems to me to be a likely explanation for the fact that no units are included on the abscissa of Figure 5 of the De Kauwe et al manuscript, defined by a combination of variables (again: the product of GPP and the square root of the vapour pressure deficit; since it would be fitting for such a group of variables to be defined and assigned a symbol, I will call it Beta). The units of Beta would necessarily include the square root of a pressure unit such as mb or Pa (equivalent to the square root of a kg m-1 s-2). My guess is that the unpleasantness of such a unit caused it to be excluded in the axis label. I would argue that Beta should be rejected altogether based on the powerful tool of dimensional analysis, which invalidates eq. (7) from the 2011 Medlyn et al. paper.**

We will add the equation underlying the analysis; the equation is given in the corrigendum to the Medlyn et al. (2011) paper, as well as many publications since, and is as follows:

$$g_s \approx 1.6(1 + \frac{g_1}{\sqrt{D}})\frac{A}{C_a}$$

Where $g_s$ is stomatal conductance (mol m$^{-2}$ s$^{-1}$), $A$ is the net assimilation rate (μmol m$^{-2}$

s$^{-1}$), $C_a$ is the $CO_2$ concentration (μmol mol$^{-1}$), $D$ is the vapour pressure deficit (kPa)

and the parameter $g_1$ (kPa$^{0.5}$) is a fitted parameter representing the sensitivity of the conductance to the assimilation rate. A full derivation for this equation is provided by

Medlyn et al. (2011). It is unclear why the reviewer thinks it is "absurd" – the equation is dimensionally correct. We agree that one should not equate different dimensions, but it is perfectly sensible to relate different dimensions: an equation may relate degrees of temperature to metres gained in elevation, for example.

Regarding the Figures: as explained in detail in the paper by Medlyn et al. (2011), it is not possible to visualise this non-linear relationship directly, but a useful approximation that allows the relationship to be visualised is to ignore the "1+" term and plot $g_s$ vs

$A/(C_a \sqrt{D})$. The slope of this relationship is then related to the parameter $g_1$. This visualisation approach is taken here but expressed in terms of transpiration. We can add further explanation of this visualisation approach to the text.

We did not include units in a similar way to other authors who have expressed water use efficiency in this fashion (e.g. Zhou, S., B. Yu, Y. Huang, and G. Wang (2014),

The effect of vapor pressure deficit on water use efficiency at the subdaily time scale,

Geophys. Res. Lett., 41, 5005–5013, doi: 10.1002/2014GL060741.). We are happy to include units on the axis of the revised figure. In addition, we will also add to the revised methods a fuller explanation for where this equation comes from.

**4. The ordinates of figures 3 and 4 are labelled with "density", a variable that**

**normally would have units such as kg m-3. Rather, I believe that what the authors**

**have plotted is a frequency of occurrence, which is a fractional, non-dimensional**

**127** **quantity that requires no units. However, since the values in figure 3 go well above**

**128** **unity, I suspect that they should be described in terms of percent (%). In any event,**

**129** **I think this needs to be clarified.**

**130** The plot is correct, and the confusion here relates to the normalisation of densities in

**131** the kernel density estimate. This is essentially the difference between probability mass

**132** functions (discrete variable) and probability density functions (continuous), the former

**133** no longer integrates to 1. We will clarify this point in our revision.

**134**

**135** **5. (This final comment may be viewed by the editor as excessively ego-centric on**

**136** **the part of the reviewer. Nonetheless I feel obligated to point it out.) I have applied**

**137** **the laws of physics to demonstrate that the paradigm underlying the definition of**

**138** **the "stomatal conductance" is fundamentally incorrect (Kowalski, Atmos. Chem.**

**139** **Phys., 17, 8177–8187, 2017), and furthermore to \*predict\* a decoupling of**

**140** **transpiration and**

**141** **photosynthesis at high temperatures. The long-standing paradigm in**

**142** **ecophysiology presupposes all transport through stomata to be diffusive in nature,**

**143** **whereas my analy- sis, based on conservation of linear momentum, shows that**

**144** **non-diffusive transport also occurs in the form of "stomatal jets". In brief, because**

**145** **the exchange of water vapour dominates surface exchange of all gases, the**

**146** **evaporation rate defines a flow velocity away from the evaporating surface and**

**147** **consequent transport of all gases away from the evaporating surface. For the**

**148** **particular case of water vapour, the analysis shows that the specific humidity**

**149** **represents the fraction of water vapour transport that is non- diffusive. Students**

**150** **of thermodynamics know that, for a saturated environment such as that supposed**

**151** **by ecophysiologists within a stomatal cavity, the specific humidity increases nearly**

**152** **exponentially as a function of temperature. Thus, at extreme temper- atures the**

**153** **role of non-diffusive transport becomes non-negligible and a decoupling is**

**154** **expected between exchanges of water vapour (whose egress is aided by non-**

**155** **diffusive transport) and carbon dioxide (whose ingress is opposed by the outgoing**

**156** **Stefan flow). At the extreme limit of the boiling point, the vapour pressure inside**

**157** **the stomatal cavity would equal the total air pressure, meaning that (1) water**

**158** **vapour would be the lone gas inside the stomatal cavity, therefore (2) no diffusion**

**could occur, and all transport would be non-diffusive (i.e., a specific humidity of 100%), and therefore (3) no photosynthesis would be possible (with no CO2 present). Since my analysis is soundly based on the laws of physics and satisfactorily explains the decoupling between photosynthesis and transpiration at high temperatures, I believe that the authors should take it into account when exploring this "previously overlooked vegetation-atmosphere feedback that may in fact dampen, rather than amplify, heat extremes". However, I hardly think it is my place to insist that other scientists cite my papers, and so must leave judgement of this matter to the editor.**

We thank the reviewer for their insight on this issue. However, we think that in order to argue for a paradigm shift ("*paradigm underlying the definition of the "stomatal conductance" is fundamentally incorrect*"), a certain weight of evidence, including measurements, will be required.

We will of course abide by the editor's decision here, but our feeling is that it would not really be appropriate to add any text regarding this work, given its relative newness and the fact that the paper referred to does not make explicit predictions for behaviour under heatwave conditions, nor even with rising temperatures. We instead would encourage the reviewer to develop their theory to make a prediction for the relative size of decoupling under heatwave conditions and test this against our published data (Drake et al. 2018, both the data and code to repeat the analysis are freely available). It might provide some empirical support for this novel and untested idea.

---

## Referee Comment (RC2) · J. Urban (Referee) · 14 Oct 2018

Recent findings on a leaf and tree level indicated that during heatwaves the photosynthesis (A) may decouple from the stomatal conductance (gs). In line with gs, transpiration (E) may increase while A does not which impacts instantaneous water use efficiency (WUE). De Kauwe's et al paper aims to extend this evidence on the ecosystem level analyzing eddy covariance data from mostly Australian forested ecosystems. Because the topic is novel and because the correlation between A and gs became central to many models the topic may be in a scope of a large audience.

Generally, the paper reads well. Data are demonstrated on figures which are mostly clear to understand. But I do not think that the research questions stated in the last

paragraph of the introduction section fit the rest of the paper. That means, the paper focuses more on changes in instantaneous WUE (A/E) that on a decoupling of the gs from A. Of course, both may be described in the same paper but the reasoning in the introduction and in a discussion as well as the structure of results should be adjusted accordingly.

Further, I suggest a few points to work on: 1. Is it E or gs which decouples from A during heatwaves? Both are interlinked but for the modeling purposes, I believe that A/gs (i.e. intrinsic WUE) is more important than A/E (i.e. instantaneous WUE). On the other hand, increase in E while A does not change or decline during the heatwave is the important issue, too. Many papers were published on E/gs which assumed stomatal regulation to maximize the A for a fixed amount of water transpired over the long time period. This idea was recently challenged (i.e. Wolf et al. 2016, PNAS; Sperry et al. 2017, PCE) and De Kauwe et al. may want to work with this evidence, should they decide to aim their paper this way. Furthermore, I do not believe that trees should keep a fixed A/E ratio in a short time (i.e. a few days of a heatwave). That said, imagine the temperature is fixed to a specific value (i.e. 25 oC) and vapor pressure deficit (VPD) increases from near zero to a couple thousand Pa (scenario unlikely to happen in nature but good to demonstrate the change in WUE). Photosynthesis would decline due to stomatal closure as a response to the increase in VPD, but the transpiration would increase. 2. Should authors want to focus more on A/gs relationship, I believe the analysis which clearly demonstrates the change in (or lack of) the response should be presented. While I do not challenge the approach of GPPxDˆ0.5 here, I do not think it is enough illustrative. Many readers, including me, are not familiar with this approach. It would be much better to demonstrate directly how A changes with a change in gs (or canopy conductance, gc). There are approaches to calculate gc from sap flow measurements (which I use). I do not know how reliable are approaches to calculate gc from eddy covariance data but if gc can be somehow derived I would be in favor of using it. 3. The timescale of the temperature vs. GPP data. Why did the authors decide to use the maximal daily temperature and compare it to the daily sum of the

[Figure]

GPP? Would not it be more appropriate to work with half an hour (or hour) resolution in both temperature and GPP? 4. What is the temperature optimum of photosynthesis for the plants in studied ecosystems? The temperature of 37oC for a part of the day may not be high enough to visibly affect the daily GPP. 5. Is there any information available how much trees and understory (grasses) contribute to the LAI and to the GPP?

---

## Referee Comment (RC3) · A. Kowalski (Referee) · 15 Oct 2018

I thank the authors for their careful replies, which nonetheless make it clear that there are issues upon which we simply disagree. Although they have not convinced me, I see little point in repeating certain arguments and prefer to leave their resolution to the discretion of the editor. Nonetheless, I wish to rebut certain points made by the authors in their response.

I find their responses to my points 1 and 2 to be essentially acceptable, although I maintain that the explicit delineation of "Conclusions" would enhance their presentation.

Regarding the issue of dimensional analysis (point 3), I feel obliged to justify my use

of the strong term "absurd", and not to back down regarding its appropriateness here. Dimensional analysis of eq. (6) of Medlyn et al. (2011) reveals that the function of incident light, f(I), must have the same units as assimilation (A), since the $CO_2$ concentration defined is a dimensionless mole fraction. In eq. (7) of Medlyn et al. (2011), the optimal stomatal conductance has the same units as f(I), such that the term in parentheses must be dimensionless. Since both the atmospheric $CO_2$ concentration (Ca) and the variable lambda are dimensionless (each expressing a mole/mole ratio), the only way that the expression under the square root sign could be dimensionless and comparable with 1 is if the constant 1.6 were to have units of m s2 kg-1 (the inverse of pressure). The authors have put this equation forth as a definition of "the theoretical expectation", and for this to be so it would be necessary that 1.6 m s2 kg-1 be some sort of universal constant, but I can find no evidence for this. Rather, from the Hari et al. (1986) paper upon which Medlyn et al. (2011) base their analyses, it appears that the factor 1.6 is simply the dimensionless ratio of the diffusivities of water vapour to $CO_2$, which derives from Graham's law (inversely proportional to the square root of the ratio of their molecular masses). Based on these analyses, I maintain my position that this "equation" is dimensionally incorrect and therefore nonsensical.

In their reply (at line 100), the authors suggest a different equation which they would add to their manuscript. Given the definitions provided in the reply, this equation is indeed dimensionally correct. However, the use of two variables with the same symbol (albeit different subscripts) and different units is unfortunate, and breaks with tradition in scientific notation. If $g_s$ is the stomatal conductance (mol m-2 s-1), then it is logical for all g variables refer to conductances with the same units (e.g., mesophyll conductance, boundary layer conductance, etc.). In equation (2) of Medlyn et al. (2011), we find $g_s$ and $g_0$ with units of mol m-2 s-1, but $g_1$ with some different units. If the units of this "key parameter" are kPaˆ0.5, as the authors propose, then eq. (2) of Medlyn et al. (2011) is also dimensionally incorrect. In short, I find the entire framework of equations to be dimensionally inconsistent (hence, "absurd"), and in any event, I suggest that a different symbol be chosen for the fitted parameter with units of kPaˆ0.5, rather than

g_1.

I do believe the only acceptable justification for excluding the units on variable axes when data are being plotted is that the variable be non-dimensional variable.

Regarding point 4, in addition to clarifying the effect of normalization on a probability function, the authors should take care to distinguish between a density (mass/volume) and a probability density. Similarly, if the probability mass function were to be presented, it would be inappropriate to label the axis with simply "mass".

Finally, I do not wish to belabor point 5, as is evidenced by the parenthetic remark with which I introduced it, and I agree that experimental evidence will likely be required to change minds and bring about a paradigm shift. However, there are some points with which I strongly disagree with the authors replies. First, the relative newness of a publication is no justification for ignoring it. Second, my paper does make specific predictions for behaviour under heatwave conditions: "Consistent with the determinants of q, as the temperature of a (saturated) stomatal environment increases, even for a constant stomatal aperture, the WUE is reduced, wresting some control over gas exchange rates from the plant." (The variable q is previously defined as the specific humidity.) Finally, the authors encourage me to develop my "theory". My paper does not present a "theory" but rather a simple derivation grounded in the basic laws of physics. It might be better described as a proof, and neither is it novel since it was derived long ago by one of the giants of classical physics (Josef Stefan).

---

## Author Comment (AC2) · 17 Oct 2018

We feel that a further response to the reviewer is required. We have separated our response into points that are germane to the paper currently under review, and some additional points regarding a previous paper that the reviewer is taking issue with (Medlyn et al. 2011).

For the paper under review:

1. We stated in the paper: "*To disentangle the potentially contributing role of D, we also estimated GPP × D$^{0.5}$ in line with the theoretical expectation of the effect of D on g$_s$*". The reviewer took this statement to indicate an equation. His comment indicates that our statement was insufficiently clear; we will clarify that our theoretical expectation is that ET is approximately proportional to GPP × D$^{0.5}$. This expectation is based on optimal stomatal theory (Lloyd et al. 1991; Medlyn et al. 2011) and is used in the "underlying WUE" approach proposed for eddy covariance data by Zhou et al. (2014, 2015). We will add citations to this latter work. Since this is a proportionality, not an equation, it cannot be dimensionally incorrect. As previously indicated, we will clarify this explanation thoroughly in the revised methods.

2. This proportionality is illustrated in Figure 5 of the paper. As we previously indicated, we agree that units should be added to x-axis of this Figure.

3. The reviewer dislikes the choice of the symbol "$g_1$" for the parameter in this equation and suggests that an alternative be chosen. While we appreciate their reasoning, we are following precedent in the stomatal literature here; a parameter $g_1$ with units other than mol m$^{-2}$ s$^{-1}$ is in common usage and has been for many years (e.g. Leuning 1990; Lloyd 1991). The nomenclature $g_1$ for a parameter with units kPa$^{0.5}$ as defined by Medlyn et al. (2011) has been used in a large number of papers since, and it would be very confusing to change this nomenclature now.

4. We can change the label "density" on Figure 4 to "Probability density".

5. We thank the reviewer for spelling out the hypothesis regarding the effect of temperature presented in their paper. Their hypothesis is that WUE should decline as temperature increases because of the change in specific humidity with temperature. This hypothesis is actually consistent with our baseline theoretical expectation that ET is proportional to GPP × D$^{0.5}$ where D increases with temperature. The hypothesis does not predict the divergence from proportionality under temperature conditions

that we are interested in, and hence we maintain that it is not directly relevant to the work presented here.

Additional points regarding Medlyn et al. (2011):

1. The reviewer queries equation 7, which reports an equation from Hari et al. (1986). Belinda Medlyn thanks the reviewer for highlighting this equation; unfortunately, the equation is missing the pressure term "P" from the numerator under the square root sign. After correcting the equation by adding this term, the equation is dimensionally correct. As this equation is not used in any further derivation, the missing term does not have any impact on the theory presented in the rest of the paper. We also note that the version of this equation presented by Hari et al. (1986) is in a slightly different form and expresses the vapour pressure difference in concentration units; it is also dimensionally correct.

2. In Medlyn et al. (2011), the parameter $g_1$ was used as a slope parameter in each of three models (Ball et al. 1987, Leuning 1995 and Medlyn et al. 2011). The parameter is different in each of these models, having a different value and different units. In retrospect, it may have been clearer in that paper to distinguish those parameters with different names (e.g. $g_{1,L}$ and $g_{1,M}$). However, that is not an issue in the current paper, which only has one equation.

3. Neither of these comments has any bearing whatsoever on the validity of the theory presented in Medlyn et al. 2011. We believe this theory has been proven to be a powerful framework for understanding stomatal behaviour, as witnessed by the large number of experimental and theoretical papers building on that work and urge the reviewer to rethink their assessment of it.

---

## Author Comment (AC3) · 18 Nov 2018

We thank the reviewer for their constructive comments and we address their various concerns below. Referee comments are highlighted in bold, with our response below in each case.

**Recent findings on a leaf and tree level indicated that during heatwaves the photosynthesis (A) may decouple from the stomatal conductance (gs). In line with gs, transpiration (E) may increase while A does not which impacts instantaneous water use efficiency (WUE). De Kauwe's et al paper aims to extend this evidence on the ecosystem level analyzing eddy covariance data from mostly Australian forested ecosystems. Because the topic is novel and because the correlation between A and gs became central to many models the topic may be in a scope of a large audience.**

**Generally, the paper reads well. Data are demonstrated on figures which are mostly clear to understand.**

We thank the reviewer for this positive summary of our work.

**But I do not think that the research questions stated in the last paragraph of the introduction section fit the rest of the paper. That means, the paper focuses more on changes in instantaneous WUE (A/E) that on a decoupling of the gs from A. Of course, both may be described in the same paper but the reasoning in the introduction and in a discussion as well as the structure of results should be adjusted accordingly.**

The reviewer is correct that whilst we do talk about a decoupling between A and $g_s$, our analysis is focussed on the response of ecosystem-scale quantities: flux-derived GPP and the flux of latent heat. However, we do not see that the question as stated, was inconsistent with our analysis - the transpiration (or latent heat flux) is in part an outcome of the leaf-level stomatal response, which we feel this was clearly articulated: "*In this paper we therefore explore eddy-covariance measurements to examine whether there is widespread field-based evidence that during heat extremes, trees decouple photosynthesis and $g_s$, leading to increased transpiration. We chose to*

*focus on wooded ecosystems as the capacity to maintain transpiration throughout a*

*heat extreme most likely requires deep soil water access and is in line with previous*

*experimental evidence from trees (Drake et al., 2018; Urban et al., 2017)*".

To further clarify this, we now also add to the above: "*In contrast to previous*

*experimental studies (e.g. Urban et al. 2017), our focus is on the ecosystem-scale and*

*so we analysed the photosynthetic decoupling between photosynthesis and*

*transpiration.*"

**Further, I suggest a few points to work on: 1. Is it E or gs which decouples from**

**A during heatwaves?**

We agree with the reviewer this is an important point to clarify and we will do so in our revised manuscript. We have changed our sub-heading in the methods 2.1 from

"*Evidence of photosynthesis-canopy conductance decoupling*" to "*Evidence of*

*photosynthesis-transpiration decoupling*".

We have also added text to this section to explain our approach: "*A number of*

*previous studies reporting photosynthetic decoupling experimentally, have focused on*

*the coupling between A and $G_s$ (Weston and Bauerle, 2007; Ameye et al. 2012; von*

*Caemmerer and Evans, 2015), as opposed to A and E (Drake et al. 2018). At the*

*ecosystem-scale (eddy-covariance), coincident measurements of $G_s$ and LE (or*

*transpiration) are rarely available. Whilst it is possible to estimate the canopy $G_s$ by*

*inverting the Penman-Monteith using measured LE, such an approach necessitates*

*additional assumptions related to the canopy boundary layer conductance (Jarvis and*

*McNaughton, 1986; De Kauwe et al. 2017), the canopy net radiation and the ground*

*heat flux (Medlyn et al. 2017). Here we avoid these assumptions by focusing our*

*analysis on the measured LE flux, as opposed to an estimate of the canopy $G_s$.*"

**Both are interlinked but for the modeling purposes, I believe that A/gs (i.e.**

**intrinsic WUE) is more important than A/E (i.e. instantaneous WUE). On the**

**other hand, increase in E while A does not change or decline during the**

**heatwave is the important issue, too.**

We agree with the reviewer; however, outside of an experimental setting we do not have access to measurements of $g_s$. Here we are seeking to examine the evidence at the ecosystem-scale and as such, our focus is on the response of E. We agree, that a decoupling of A/$g_s$ may not translate to A/E at the canopy/ecosystem-scale due to the level of control stomata have on transpiration ("decoupling", Jarvis and McNaughton, 1986) and environmental drivers (net radiation, wind speed, VPD). In our revised methods (see above) and discussion text we explain this point more fully.

The point the reviewer highlights speaks to the novelty of our approach, which considers responses at the ecosystem-scale and attempts to contextualise previous experimental work (e.g. Drake et al. 2018 and Urban et al. 2017).

**Many papers were published on E/gs which assumed stomatal regulation to maximize the A for a fixed amount of water transpired over the long time period. This idea was recently challenged (i.e. Wolf et al. 2016, PNAS; Sperry et al. 2017, PCE) and De Kauwe et al. may want to work with this evidence, should they decide to aim their paper this way.**

In our revised discussion (4.2 Implications for models), we have addressed the point raised by the reviewer: "*The implications for modelling studies that focus on heat extremes are clear, particularly for studies in Australia. None of the current generation of land surface models have the capacity to decouple transpiration from the down-regulation of photosynthesis with increasing temperature. Instead models assume photosynthesis and $g_s$ (and consequently transpiration) remain coupled at all times. As a result, climate models will underestimate the capacity of the vegetation to dampen heat extremes in simulations for Australia. This is also true of more sophisticated plant hydraulic models (Williams et al. 2001) and profit-maximisation approaches (Wolf et al. 2015; Sperry et al. 2016) that hypothesise the cost of water is not fixed in time, but instead increases with increasing water stress. For these latter approaches to account for a photosynthetic decoupling they would need to prioritise maintaining an optimum canopy temperature above a net carbon gain. However, mechanisms to capture this within models should likely wait for further supporting evidence of photosynthetic decoupling.*"

**Furthermore, I do not believe that trees should keep a fixed A/E ratio in a short time (i.e. a few days of a heatwave). That said, imagine the temperature is fixed to a specific value (i.e. 25 oC) and vapor pressure deficit (VPD) increases from near zero to a couple thousand Pa (scenario unlikely to happen in nature but good to demonstrate the change in WUE). Photosynthesis would decline due to stomatal closure as a response to the increase in VPD, but the transpiration would increase.**

This is why we also analysed the eddy-covariance data from the perspective of WUE, to attempt to disentangle any decoupling from the response to increasing VPD.

**2. Should authors want to focus more on A/gs relationship, I believe the analysis which clearly demonstrates the change in (or lack of) the response should be presented.**

As mentioned above, it is not possible to show the $A/g_s$ relationship from eddy-covariance data. To do so would require inverting the Penman-Monteith equation from measured LE flux. Whilst this approach has been used, it requires a series of assumptions related to the canopy aerodynamic conductance, it is far clearer to analysis the measured flux. See added text above.

**While I do not challenge the approach of GPPxDˆ0.5 here, I do not think it is enough illustrative. Many readers, including me, are not familiar with this approach.**

We agree that we were not clear enough in our explanation of this approach, a point that the other reviewer also highlighted. In our revised methods, we now explain why we took this approach: "*As temperature increases, vapour pressure deficit (D) also increases, which will drive an increase in LE unless there is stomatal closure, but this effect is unrelated to the decoupling mechanism we seek to find. To disentangle the potentially contributing role of D, we also explored these data based on the theoretical*

*expectation (Lloyd et al. 1991; Medlyn et al. 2011; Zhou et al. 2014) that transpiration (E) is approximately proportional to GPP $\times$ $D^{0.5}$ (g C $kPa^{0.5}$ $m^{-2}$ $d^{-1}$; Eqn. 7). This expectation is based the idea of optimal stomatal behaviour proposed by Cowan and Farquhar (1977) that stomata should be regulated so as to maximise photosynthetic carbon gain less the cost of transpiration. Medlyn et al. (2011) derived the optimal stomatal behaviour as:*

$$G_s = 1.6\left(1 + \frac{g_1}{\sqrt{D}}\right)\frac{A}{C_a} \tag{1}$$

*where $G_s$ is canopy stomatal conductance to $CO_2$ (mol $m^{-2}$ $s^{-1}$), A is the net assimilation rate ($\mu$mol $m^{-2}$ $s^{-1}$), $C_a$ is the ambient atmospheric $CO_2$ concentration ($\mu$mol $mol^{-1}$), D is the vapour pressure deficit (kPa), the parameter $g_1$ ($kPa^{0.5}$) is a fitted parameter representing the sensitivity of the conductance to the assimilation rate and the factor 1.6 is the ratio of diffusivity of water to $CO_2$ in air. Assuming that transpiration is largely controlled by conductance, this relationship can be rearranged to show that water-use efficiency (A/E) is approximately proportional to $1/\sqrt{D}$. This dependence has been remarked by many authors (e.g. Lloyd et al. 1991, Katul et al. 2009). Based on this dependence, Zhou et al. (2014, 2015) proposed an "underlying water-use efficiency" (uWUE) for eddy covariance data:*

$$uWUE \approx \frac{GPP\sqrt{D}}{E} \tag{2}$$

*Zhou et al. (2014) argued that the $D^{0.5}$ term provided a better linear relationship between GPP and E. Thus, to probe the effect of D, we focused on heatwaves (i.e. approach 2) and plotted LE expressed as evapotranspiration (mm $day^{-1}$), as a function of GPP$\times D^{0.5}$."*

**It would be much better to demonstrate directly how A changes with a change in gs (or canopy conductance, gc). There are approaches to calculate gc from sap flow measurements (which I use). I do not know how reliable are approaches to calculate gc from eddy covariance data but if gc can be somehow derived I would be in favor of using it.**

See response above.

**3. The timescale of the temperature vs. GPP data. Why did the authors decide to use the maximal daily temperature and compare it to the daily sum of the GPP? Would not it be more appropriate to work with half an hour (or hour) resolution in both temperature and GPP?**

The suggested approach is of course a viable analysis framework; however, it would increase the time-resolution (and so the noise in the data) without necessarily adding any additional insight. Our approach analysed multiple heat-extreme events, across multiple site, this would not be possible (or would be harder) if we disaggregated this into diurnal, 4-day events. Here, we are seeking to see the broader patterns at behaviour at the ecosystem-scale.

**4. What is the temperature optimum of photosynthesis for the plants in studied ecosystems? The temperature of 37oC for a part of the day may not be high enough to visibly affect the daily GPP. 5.**

The temperature optima for leaf and canopy photosynthesis in Eucalypts in southern Australia are well below 30 degrees (see Duursma et al. 2014; Drake et al. 2016; Kumarathunge et al. in review), suggesting that days above 37 degrees should induce a decline in GPP. We also analysed heatwave events (defined as least three consecutive days where the maximum daily temperature exceeded 35°C).

We have addressed this point in our new discussion sub-section (4.1 Why did we not find supporting evidence for ecosystem-scale photosynthetic decoupling?), specifically: "*One could ask whether our analysis considered hot enough temperatures (> 37 °C) to trigger a photosynthetic decoupling mechanism. For example, during an imposed heatwave, Ameye et al. (2012) probed the decoupling mechanism at daily maximum temperatures between 47 and 53°C. Similarly, Zhu et al. (2018) found that most of the 62 species sampled across Australia exhibited maximum critical temperatures near 50°C. However, the temperature optima for leaf and canopy photosynthesis in Eucalypts in southern Australia are well below 30 degrees (see Duursma et al. 2014; Drake et al. 2016; Kumarathunge et al. in review), suggesting that days above 37°C should induce a decline in GPP. Our analysis also*

*included events with daily maximum temperatures of greater than 40°C and*
*consecutive heatwave days > 35°C. Therefore, we would argue that insufficiently*
*high temperatures are unlikely to explain why we did not see clear evidence when*
*looking at eddy covariance data.*"

**Is there any information available how much trees and understory (grasses)**
**contribute to the LAI and to the GPP?**

Across all of these flux sites we analysed, the simple answer is no. We have now added
a statement on this issue of leaf area adjustment to our new discussion: "*Finally,*
*although Drake et al. (2018) did not find evidence of increased litterfall during their*
*heatwave experiment, it is of course possible that at our sites, there was some reduction*
*in leaf area in response to high extremes. Any leaf area reduction would in turn reduce*
*transpiration and thus may offset ecosystem-scale estimates of a photosynthetic*
*decoupling.*"

---

## Author Comment (AC4) · 18 Nov 2018

We thank the reviewer for their constructive comments and we address their various concerns below. Referee comments are highlighted in bold, with our response below in each case. We note that we made two earlier responses to the reviewer during revision, this response now incorporates the key points of those interactions to make things easier for the editor.

**Prompted by recent observations from chamber measurements of a decoupling be- tween photosynthesis and transpiration at high temperatures, De Kauwe and col- leagues examine eddy covariance flux data to see whether such decoupling can be observed at the ecosystem scale. To my mind, this manuscript suffers from several important inadequacies, and requires major revision before it would be acceptable for publication. Anticipating that some of my criticisms will be viewed as controversial, I will nonetheless lay them all out, so that the editor can determine which (if any) deserve to be taken into consideration:**

**1. Both Tier-1 FLUXNET2015 data and OzFlux data suffer doubts regarding their validity due to their persistent failure to demonstrate conformity with the principle of energy conservation (i.e., to close the surface energy budget). Although it might be going too far to say that it is inappropriate to download and analyze such data as the authors have done, neither do I think it is correct for this issue to be neglected entirely. Specifically, I am not aware that anyone has looked at the effect of heat waves on the energy balance closure, but this would certainly seem to be germane to the scientific questions that the authors are posing in the context of dataset validity. Also, although the FLUXNET2015 database includes a GPP variable, this is not measured by flux towers and the procedure from which it is inferred is of dubious validity during conditions of extreme heat stress. Given that the authors are attempting to tease out subtle temperature dependencies of GPP (which is not measured directly) and LE (which fails energy conservation checks), it seems inappropriate to me that such issues are not mentioned at all in this paper.**

We appreciate the Reviewers concerns on this issue.

We note in response to their statement about GPP that on page 6 of our original submission that we stated: *"Our analysis also relies on GPP which is not directly*

*observed but is instead modelled using assumptions related to the extrapolation of night-time respiration (ER) and measured net ecosystem exchange. It is debatable whether these assumptions hold at very high temperatures, and examining these modelled GPP estimates at high temperatures warrants further investigation particularly as researchers leverage these data to explore the responses of the vegetation to temperature extremes*."

In our revised discussion we have more fully addressed this concern: "*Our approach relies on GPP which is not directly observed but is instead modelled using assumptions related to the extrapolation of night-time respiration and measured net ecosystem exchange. It is debatable whether these assumptions hold at very high temperatures, and examining these modelled GPP estimates at high temperatures warrants further investigation, particularly as researchers leverage these data to explore the responses of the vegetation to temperature extremes. Eddy-covariance data are also known to have issues closing the energy balance (see Wohlfahrt et al. 2009, for a detailed discussion), which may introduce errors into the LE flux. For the seven Australian flux sites that make up the majority of our analysis, we calculated the ratio of the sum of latent and sensible heat fluxes to the sum of the net radiation and ground heat flux, finding on average a ~17% imbalance in the ratio (minimum=30%; maximum=7%). Importantly however, we did not find any difference in this imbalance in heatwave vs. non- heatwave days. Despite these limitations, FLUXNET eddy covariance flux measurements still present our best ecosystem-scale estimates of vegetation responses to heat extremes and have been widely analysed to address these types of questions (Ciais et al. 2005; Teuling et al. 2010; Wolf et al. 2013; von Buttlar et al. 2018; Flach et al. 2018).*"

**2. The paper draws no concrete conclusions, partly I think because the organisation of the manuscript is below standard.**

We would disagree with this interpretation. We draw no concrete conclusions because the data do not allow us to do so. In our paper we tested whether a photosynthetic decoupling mechanism identified in whole-tree chamber experiments (e.g. Drake et al. 2018, Global Change Biology), as well as other leaf-level experiments, was present at the ecosystem scale. As our results demonstrate, outside of the experimental

environment, it is difficult to isolate such a mechanism. We did not find strong support for the original experimental result. However, absence of evidence is not evidence of absence and, given the caveats attached to the data, more concrete conclusions would be unwarranted. Instead, we discussed the need for new field-based studies to tackle this issue further. Although we are unable to draw concrete conclusions, we nonetheless believe the analysis is worth publishing as this is the first study to test for photosynthetic decoupling at an ecosystem scale and as such, discuss the associated uncertainties. Our revised Discussion section also includes a route forward section, which may help satisfy the reviewer on the merit of the study.

**The paper contains about 1 page of introduction, 1.5 pages of methods, and 2.5 pages of "Results and discussion" to which will be added five figures and a table. This last section makes for difficult reading, in part because the authors appear to make little effort to distinguish between the facts and their inter- pretations thereof. Furthermore, the paper contains no equations whatsoever, despite the fact that the authors plot a variable (the product of GPP and the square root of the vapour pressure deficit) whose grouping cannot be justified (see comment number 3 below). All of these structural shortcomings make it particularly difficult for the reader to extract and evaluate the underlying message of the manuscript. I believe that the paper would be much better organised with a classical structure of 1. Introduction 2. Methods 3. Results 4. Discussion & 5. Conclusions.**

We have now reorganised our manuscript as the reviewer suggested, adding an improved Methods and new Discussion and Conclusion sections.

**3. According to the abstract, an important aspect of the paper addresses "the role of vapour pressure deficit" (D). The authors describe this in terms of the "theoretical expectation of the effect of D on $g_s$" (page 3, line 27), citing previous works in this regard. Although not explicitly appearing in this manuscript, the "equation" underlying this idea is eq. (7) from the 2011 paper by Medlyn et al., which is demonstrably in- correct. One of the major contributions to science of Joseph Fourier is the criterion of "dimensional homogeneity", which states that only quantities with the same dimen- sion can be compared, equated, added or**

**subtracted. An obvious example would be the ridiculous statement that one kilometer is greater than one second. At the risk of sounding harsh, I must point out that equation (7) of the Medlyn et al. (2011) paper is equally absurd, and should not be considered as a "theoretical expectation". This ab- surdity seems to me to be a likely explanation for the fact that no units are included on the abscissa of Figure 5 of the De Kauwe et al manuscript, defined by a combination of variables (again: the product of GPP and the square root of the vapour pressure deficit; since it would be fitting for such a group of variables to be defined and assigned a symbol, I will call it Beta). The units of Beta would necessarily include the square root of a pressure unit such as mb or Pa (equivalent to the square root of a kg m-1 s-2). My guess is that the unpleasantness of such a unit caused it to be excluded in the axis label. I would argue that Beta should be rejected altogether based on the powerful tool of dimensional analysis, which invalidates eq. (7) from the 2011 Medlyn et al. paper.**

We have now clearly explained the theory that supports our analysis: "*As temperature increases, vapour pressure deficit (D) also increases, which will drive an increase in LE unless there is stomatal closure, but this effect is unrelated to the decoupling mechanism we seek to find. To disentangle the potentially contributing role of D, we also explored these data based on the theoretical expectation (Lloyd et al. 1991; Medlyn et al. 2011; Zhou et al. 2014) that transpiration (E) is approximately proportional to GPP $\times D^{0.5}$ (g C $kPa^{0.5}$ $m^{-2}$ $d^{-1}$; Eqn. 7). This expectation is based the idea of optimal stomatal behaviour proposed by Cowan and Farquhar (1977) that stomata should be regulated so as to maximise photosynthetic carbon gain less the cost of transpiration. Medlyn et al. (2011) derived the optimal stomatal behaviour as:*

$$G_s = 1.6\left(1 + \frac{g_1}{\sqrt{D}}\right)\frac{A}{C_a} \tag{1}$$

*where $G_s$ is canopy stomatal conductance to $CO_2$ (mol $m^{-2}$ $s^{-1}$), A is the net assimilation rate ($\mu mol$ $m^{-2}$ $s^{-1}$), $C_a$ is the ambient atmospheric $CO_2$ concentration ($\mu mol$ $mol^{-1}$), D is the vapour pressure deficit (kPa), the parameter $g_1$ ($kPa^{0.5}$) is a fitted parameter representing the sensitivity of the conductance to the assimilation rate and the factor 1.6 is the ratio of diffusivity of water to $CO_2$ in air. Assuming that transpiration is largely controlled by conductance, this relationship can be rearranged to show that water-use efficiency (A/E) is approximately proportional to $1/\sqrt{D}$. This dependence has been remarked by many authors (e.g. Lloyd et al. 1991, Katul et al. 2009). Based on*

*this dependence, Zhou et al. (2014, 2015) proposed an "underlying water-use efficiency" (uWUE) for eddy covariance data:*

$$uWUE \approx \frac{GPP\sqrt{D}}{E} \qquad (2)$$

*Zhou et al. (2014) argued that the $D^{0.5}$ term provided a better linear relationship between GPP and E. Thus, to probe the effect of D, we focused on heatwaves (i.e. approach 2) and plotted LE expressed as evapotranspiration (mm day$^{-1}$), as a function of GPP$\times D^{0.5}$."*

Both of our earlier responses to reviewer argued that there was in fact no problem with units, rather our original submission was simply not clear enough. We hope that our revised text will now satisfy the reviewer that there are no further issues. We refer the editor to earlier responses on this issue.

We have also added the requested units to the figure labels.

**4. The ordinates of figures 3 and 4 are labelled with "density", a variable that normally would have units such as kg m-3. Rather, I believe that what the authors have plotted is a frequency of occurrence, which is a fractional, non-dimensional quantity that requires no units. However, since the values in figure 3 go well above unity, I suspect that they should be described in terms of percent (%). In any event, I think this needs to be clarified.**

The plot is correct, and the confusion here relates to the normalisation of densities in the kernel density estimate. This is essentially the difference between probability mass functions (discrete variable) and probability density functions (continuous), the former no longer integrates to 1. We have now added "Probability density" to the figure label and added an interpretation sentence to each of the figure captions.

**5. (This final comment may be viewed by the editor as excessively ego-centric on the part of the reviewer. Nonetheless I feel obligated to point it out.) I have applied the laws of physics to demonstrate that the paradigm underlying the definition of the "stomatal conductance" is fundamentally incorrect (Kowalski, Atmos. Chem.**

**Phys., 17, 8177–8187, 2017), and furthermore to \*predict\* a decoupling of transpiration and**

**photosynthesis at high temperatures. The long-standing paradigm in ecophysiology presupposes all transport through stomata to be diffusive in nature, whereas my analy- sis, based on conservation of linear momentum, shows that non-diffusive transport also occurs in the form of "stomatal jets". In brief, because the exchange of water vapour dominates surface exchange of all gases, the evaporation rate defines a flow velocity away from the evaporating surface and consequent transport of all gases away from the evaporating surface. For the particular case of water vapour, the analysis shows that the specific humidity represents the fraction of water vapour transport that is non- diffusive. Students of thermodynamics know that, for a saturated environment such as that supposed by ecophysiologists within a stomatal cavity, the specific humidity increases nearly exponentially as a function of temperature. Thus, at extreme temper- atures the role of non-diffusive transport becomes non-negligible and a decoupling is expected between exchanges of water vapour (whose egress is aided by non-diffusive transport) and carbon dioxide (whose ingress is opposed by the outgoing Stefan flow). At the extreme limit of the boiling point, the vapour pressure inside the stomatal cavity would equal the total air pressure, meaning that (1) water vapour would be the lone gas inside the stomatal cavity, therefore (2) no diffusion could occur, and all transport would be non-diffusive (i.e., a specific humidity of 100%), and therefore (3) no photosynthesis would be possible (with no $CO_2$ present). Since my analysis is soundly based on the laws of physics and satisfactorily explains the decoupling between photosynthesis and transpiration at high temperatures, I believe that the authors should take it into account when exploring this "previously overlooked vegetation-atmosphere feedback that may in fact dampen, rather than amplify, heat extremes". However, I hardly think it is my place to insist that other scientists cite my papers, and so must leave judgement of this matter to the editor.**

We thank the reviewer for their insight on this issue. Despite our back and forth discussion on this topic, we still maintain that that in order to argue for a paradigm shift ("*paradigm underlying the definition of the "stomatal conductance" is fundamentally incorrect*"), a certain weight of evidence, including measurements, will be required.

195    We further thank the reviewer for spelling out the hypothesis regarding the effect of

196    temperature presented in their paper. Their hypothesis is that WUE should decline as

197    temperature increases because of the change in specific humidity with temperature.

198    This hypothesis is actually consistent with our baseline theoretical expectation that E

199    is proportional to $GPP \times D^{0.5}$ where D increases with temperature. The hypothesis

200    does not predict the divergence from proportionality under temperature conditions

201    that we are interested in, and hence we maintain that it is not directly relevant to the

202    work presented here. However, if the editor feels we should refer to this work, we will

203    of course abide by their decision here.

---

## Author Response (AR2)

Dear Professor Yakir,

I am writing to resubmit our manuscript, now titled: "*Examining the evidence for decoupling between photosynthesis and transpiration during heat extremes*" for consideration for publication in Biogeosciences.

We have addressed each of the ten points outlined by the Editor, we thank the Editor for their constructive suggestions.

Thank you for your consideration.

Yours Faithfully,

Martin G. De Kauwe (on behalf of all the authors)

We thank the additional reviewer and the Associate Editor for summarising and
 highlighting their outstanding concerns. They raised a number of constructive points,
 which we have used to help craft our revised manuscript. The Associate Editor's
 comments are highlighted in bold, with our response below in each case.

5

6 As stated at the outset, the paper deals with an interesting and important issue of 7 decoupling A vs T in response to extreme events, specifically during heat waves, 8 and should be appropriate for publication in BG. The paper was revised and 9 improved, mainly in reorganizing and adding a useful Discussion section 10 (although the Results section is still "results and discussion"). And I do agree with 11 the notion that inconclusive results in a good study can still be valuable. But with 12 some controversy and with some partial responses to comments, the paper was 13 sent out for review and check by additional expert reviewer. Indeed, the reviewer 14 noted that he was "somewhat disappointed with some of the replies to the 15 reviewer's comments..." I therefore try below to highlight a few of the points I 16 noted (I did not extensively review the paper), and recommend additional 17 revisions to further improve the paper before final publication.

We thank the Editor for this positive summary. We acknowledge we could have been more thorough on some of our responses and we think that this additional round of reviews has now led us to provide that detail requested.

21

1. The title could be improved as suggested to better reflect what is repeatedly stated as the topic—"trees decouple photosynthesis and gs..."; and "we analysed the photosynthetic decoupling between photosynthesis and transpiration..." (the current title seems to point to the VPD driven response which the paper wants to extend).

As suggested, we have changed the title to: "Examining the evidence for decouplingbetween photosynthesis and transpiration during heat extremes".

29

30 2. On the same token, one Ref noted the importance of considering in more
31 detail conductance. It seems that A vs E decoupling would indeed require g

response, and it is somewhat odd that while the paper has no problems making assumptions regarding LE and GPP, it would not attempt to estimate or discuss any form of G that is often obtained at the canopy scale, at least to some extent without extensive modeling (e.g. from VPD\*/LE when LE is argued to reliably reflect T and temperature records are discussed in detail).

37 Ultimately, we would suggest to the Editor that there is no "correct" way to do this 38 and it is a clearly a choice where you make your assumptions. For example, in the 39 Tatarinov study that the Editor cites below, the authors infer canopy conductance by 40 assuming a perfectly coupled boundary layer. Recent synthesis work by De Kauwe et 41 al. (2017), hints this is not the case, even in pine stands. The assumption we make 42 about ET and soil evaporation (we extend the text on this below) would, we suggest, 43 be far less important than the role of the boundary layer. Crucially, whether we used 44 the measured LE flux or inferred the canopy conductance, it would not change what 45 we have learnt in this manuscript about ecosystem-scale photosynthetic decoupling -46 it amounts to the same thing. We would also highlight that our analysis is in line with 47 the approach used in the recent Drake et al. (2018) paper which has been already cited 48 16 times and was one of Global Change Biology's most downloaded papers in 2018. 49 Thus, we felt our previously revised text was a fair reflection of these choices, noting 50 that we have now added further text to address the soil evaporation assumption (see 51 below).

52

53 De Kauwe, M. G., Medlyn, B. E., Knauer, J., and Williams, C. A.: Ideas and

54 perspectives: how coupled is the vegetation to the boundary layer? Biogeosciences,

55 14, 4435-4453, https://doi.org/10.5194/bg-14-4435-2017, 2017.

56

57 3. Similarly, substituting LE for T, and indirectly estimated GPP are now 58 better noted with references, but the implications to the current study, such as to 59 the slope vs temp are not discussed. (e.g. understory and LAI issue are commented 60 by two Refs).

Thanks for the positive comments. To help resolve the "implications" comment we
have now extended the paragraph in the discussion where we raised the point about
LAI to deal with the issue raised by the Editor: "*Although Drake et al. (2018) did not*

find evidence of increased litterfall during their heatwave experiment, it is of course possible that across the FLUXNET sites we considered, there was some reduction in leaf area in response to high temperature extremes. However, any leaf area reduction would reduce both transpiration and photosynthesis and thus, we think it is unlikely to affect ecosystem-scale estimates of a photosynthetic decoupling. Nevertheless, future flux-based experiments may consider also using leaf litter traps at sites to allow researchers to separate out this effect and confirm this assumption."

71

72 We have also extended the text to discuss the implications for understory fluxes. Firstly 73 in the methods: "...or in the two days prior to a heat event in the eddy covariance data 74 (Dekker et al., 2001; Law et al., 2002; Groenendijk et al., 2011; Keenan et al., 2013; 75 Dekker et al. 2016; De Kauwe et al. 2017; Knauer et al. 2017; Medlyn et al. 2017) as 76 this could bias the LE flux by leading to an increase in LE not associated with the 77 mechanism we wished to identify (i.e. due to soil/canopy evaporation). Knauer et al. 78 (2017) is the only study to have explicitly tested the impact of assuming that two days 79 following a rainfall event, the LE flux can be assumed to dominated by transpiration. 80 Across six FLUXNET sites, they found between a 9% and 19% change in estimates of 81 the slope parameter of the optimal stomatal parameter  $(g_1; Medlyn et al. 2017)$  with 82 increasing time since the last rainfall event beyond 48 hours (out to 240 hours). 83 However, their analysis did not account for the potential confounding effect that as they 84 screened a greater number of hours following rainfall, the number of samples used to 85 estimate the  $g_1$  parameter was also reduced, which would increase the error in 86 estimates of the model parameter. Given both the high temperatures considered in our analysis framework and the length of the period after screening for rain (at least three 87 88 days), we would expect the impact of soil evaporation to be a minor consideration."

89

90 And also in the Discussion: "Finally, throughout our manuscript we have treated the 91 measured LE flux as interchangeable for the transpiration flux (i.e. ignoring any 92 potential role of soil and or canopy evaporation – see Methods 2.2). Strictly, if soil 93 and/or canopy evaporation fluxes were not zero, the signal that we have analysed would 94 contain a contribution that is not directly under the plants control and so could not be

95 affected (directly) by any photosynthetic decoupling. Whilst we cannot rule out such a 96 contribution we expect it to be unlikely to be a significant factor at play in our results. 97 Screening the eddy covariance timeseries for the two days following observed rain 98 events follows a widely used strategy in eddy covariance studies (Dekker et al., 2001; 99 Law et al., 2002; Groenendijk et al., 2011; Keenan et al., 2013; Dekker et al. 2016; De 100 Kauwe et al. 2017; Knauer et al. 2017; Medlyn et al. 2017). Moreover, our analysis 101 also considers heat extremes that last for at least three further days. Thus, after five 102 days (two days prior to event must also have been rain free), in temperatures exceeding 103 *30°C, we think it likely that the latent heat flux will be dominated by transpiration.*"

104

**4. As suggested the authors added more explanation of the theory/background leading to their analysis, but it's hidden in the Methods while it fits better up front in the Intro.**

We have now reorganised the text – introducing the three relevant paragraphs from the methods into the end of the introduction, whilst leaving the text which related directly to the methodology of our approach in the methods.

111

5. There were also related arguments on the physical basis of some of the complicated 'units' and in addition to noting these "units" it will help readers to better clarify that indeed, there are some none-physical parameters that derived from 'relationships' or 'correlation' and as such may not require formal units.

116 We have added: "We note for the interested reader tracing the development of the

117 optimal stomatal theory through the cited publications, that equation 7 in Medlyn et al.

118 (2011) is missing a pressure (P) term in the numerator (under the square root sign),

119 which ensures the equation is dimensionally correct. However, the equation is not used

120 *in any further derivation in Medlyn et el. (2011) and so the missing term does not have*

121 *any impact on the theory presented in the rest of that paper.*"

122

123 6. In the context of discussing the theoretical basis of a core issue such as
124 unexpected conductance or transpiration response, and considering that Kowalski
125 happened to be a reviewer (fortuitously), I do think it's appropriate to cite his less

126 conventional study, especially as the authors note it adds another perspective rather than a contradiction to the results. The doubts regarding leaf internal 127 128 vapor pressure are at least as intriguing as the decoupling issue. The Kowalski 129 paper may in fact be relevant together with another, contrasting, paper recently 130 out (Cernusak et al 2018) on reduced e inside leaves at high VPD. Maybe less of 131 concern are the other suggested references of Vesala et al., and of Eder et al., 132 although it's surprising that the later one is rejected because temperature at the 133 site was not found (there are dozens of papers on that particular site, including at 134 least two specifically on heat-wave response in pine trees; Tatarinov et al., 2015; 135 Wohlfahrt et al., 2018).

136 We now cite the Tatarinov et al. study (see response to point 8), the Eder et al. study 137 when we discuss the issue of energy balance closure and have added a paragraph to the 138 discussion to cite the Kowalski et al. and Cernusak et al. studies: "As the background 139 climate warms with associated changes in the intensity and frequency of heat extremes, 140 there is a growing interest in the degree to which leaf temperature affects, and is 141 affected by, the physiological response of plants. The potential for plants to use a 142 photosynthetic decoupling mechanism to apparently regulate leaf temperatures is one 143 emerging aspect of this interplay between plant physiology and temperature. Other 144 studies are currently questioning other widely-accepted notions about stomatal 145 regulation. For example, Cernusak et al. (2018) recently examined the near universal 146 assumption that vapour pressure inside a leaf remains saturated in all conditions. They 147 found in two conifer species that, under moderate to high D, this assumption was 148 invalid leading to a bias in the calculated gs. Similarly, Kowalski et al. (2017) have recently questioned the paradigm that all transport through stomata is diffusive, 149 150 instead invoking the concept of non-diffusive stomatal jets. However, neither of these 151 papers provides a mechanism by which stomatal closure would be decoupled from 152 photosynthesis. Further plant physiological studies are required to identify this 153 mechanism."

154

155 7. A Ref point regarding the data time scales is partially addressed in the
156 Response and not at all in the text. Seems valid to ask why daily sum GPP could

**not be coupled, for example, with daily mean temperatures, or max temp relatedto midday GPP?**

159 We agree with the Editor, we should have commented on this, we have extended the 160 text now: "For each of these events we recorded the maximum daytime temperature, 161 the mean daytime (6am - 8 pm) latent heat flux (LE), and the daytime summed gross 162 primary productivity (GPP). Although we chose to compare mean daytime LE and the 163 summed daytime GPP with the maximum daytime temperature, there are of course 164 alternative analysis approaches. We chose our approach as an appropriate trade off in 165 time resolution that facilitated us to consider several heat-extreme events, across 166 multiple sites. This allowed us to see the broader patterns of behaviour at the 167 ecosystem-scale. Had we considered analysing the raw 30-minute data for example, we 168 *felt that interpretation of the underlying behaviour would been made considerably more* 169 difficult due to the increased time frequency and inherent noise in these data. A further 170 alternative analysis approach would have been to compare the maximum or daily mean 171 temperature with the midday GPP and LE fluxes; however, we felt such an approach 172 could miss interesting morning and afternoon responses which may result directly from 173 the temperature extremes but not be present in the midday observation."

174

175 8. Finally, in the context of the discussion, it might be relevant to discuss the 176 broader point that the expectation of decoupling that involves enhanced T, is 177 relevant to conditions of very high temp associated with non-limiting water 178 supply, which may not be the "norm"...

179 We agree and we have added to the discussion: "A number of the previous studies that 180 showed photosynthetic decoupling experimentally were carried out on well-watered 181 plants (Ameye et al. 2012; Urban et al. 2017). Thus, one interpretation of our results 182 is simply that root-zone soil moisture was limiting any photosynthetic decoupling. In 183 Drake et al. (2017), irrigation of the whole-tree chambers was withheld for the month 184 prior to the heatwave experiment, thus a more nuanced interpretation may be that a 185 photosynthetic decoupling mechanism requires access to soil moisture from deeper in 186 the profile (perhaps associated with access to groundwater). Without data throughout 187 the root-zone profile across the flux sites we cannot rule out this explanation. Our 188 results did show tentative evidence consistent with this explanation; we found a small 189 decreased in the number of positive slopes (i.e. increased LE) towards the end of the 190 summer (Fig S3), which may reflect reduced soil water availability to sustain 191 transpiration. Using sap flow data, Tatarinov et al. (2015) found a ~60% decrease in 192 canopy conductance, an approximately halving of daytime GPP, but little change in ET 193 during spring heat waves ('hamsin') in a 50-year-old Alepp pine forest located at the 194 edge of the Negev desert. The observed responses during these Mediterranean heat 195 extremes are consistent with a photosynthetic decoupling although in their study, we 196 note that the authors attributed these differences in behavior to the relative influence 197 of D and soil moisture availability."

198

199 9. More technically, Figure 1, and probably Fig 2, are ineffective in showing 200 slopes or trends, as the scale in panels a to f seem to be dictated by that of the 201 unusually high values of panel g. Without adjusting the scale, it seems not much 202 point in displaying them. In the captions, the term slope is used and it would help 203 to more clearly indicate what slope.

204 We agree with the Editor and have now used individual y-ranges for each subplot,

which more clearly shows site variations. We have also adjusted the caption to

206 indicate that we aren't showing the slope: "Note in both cases, we are not showing the

207 fitted slopes, we are simply using this approach to identify stronger positive or

208 negative trends in these data (see methods)."

209

**10.** The new Discussion and conclusion is a good addition, but the separation**

- 211 of the sub-sections of "Route forward" and "Conclusions" seems excessive.
- 212 We agree and we have combined these sections.

**Examining the evidence for decoupling between photosynthesis and transpiration during heat extremes**

Martin G. De Kauwe1, Belinda E. Medlyn2, Andrew J. Pitman1, John E. Drake3, Anna Ukkola4, Anne Griebel2, Elise Pendall2, Suzanne Prober5 and Michael Roderick4

[revised manuscript text omitted]

| 10     Formatted: Forn: 10 pt, Bold, English (UK)       10     Formatted: Normal (Web), Space Before: 0 pt, After: 0 pt,
Line spacing: 1.5 lines       15     Formatted: Forn: Bold, English (UK)       20     Formatted: Forn: Bold, English (UK)       21     Formatted: Forn: Bold, English (UK)       22     Formatted: Forn: Bold, English (UK) | 5  |  <li>Yu, GR., Wen, XF., Sun, XM., Tanner, B. D., Lee, X. and Chen, JY.: Overview of Chinaflux and evaluation of its eddy covariance measurement. Agricultural and Forest Meteorology, 137, 125–137, 2006.</li> <li>Zhu L, Bloomfield KJ, Hocart CH, et al.: Plasticity of photosynthetic heat tolerance in plants adapted to thermally contrasting biomes. Plant, Cell & Environment, 18, 1251-1262, 2018.</li>  |        |                                                                                                                                    |
|---------------------------------------------------------------------------------------------------------------------------------------------------------------------------------------------------------------------------------------------------------------------------------------------------------------------------------------------------------|----|-------------------------------------------------------------------------------------------------------------------------------------------------------------------------------------------------------------------------------------------------------------------------------------------------------------------------------------------------------------------------------------------------------------------------------|--------|------------------------------------------------------------------------------------------------------------------------------------|
| 15       Formatted: Font: Bold, English (UK)         20       25         30       Formatted: Judified         Deleted: 21       Pormatted: Page Number         Formatted: Page Number       Formatted: Page Number                                                                                                                                      | 10 | •                                                                                                                                                                                                                                                                                                                                                                                                                             | $\sim$ | Formatted: Font: 10 pt, Bold, English (UK)
Line spacing: 1.5 lines |
| 20
25
30
Formatted: Page Number                                                                                                                                                                                                                     | 15 | A                                                                                                                                                                                                                                                                                                                                                                                                                             |        | Formatted: Font: Bold, English (UK)                                                                                                |
| 25
30
Formatted: Page Number                                                                                                                                                                                                                                                     | 20 |                                                                                                                                                                                                                                                                                                                                                                                                                               |        |                                                                                                                                    |
| 30
Formatted: Page Number                                                                                                                                                                                                                                                           | 25 |                                                                                                                                                                                                                                                                                                                                                                                                                               |        |                                                                                                                                    |
| Formatted: Justified         Deleted: 21         Formatted: Page Number         Formatted: Page Number                                                                                                                                                                                                                                                  | 30 |                                                                                                                                                                                                                                                                                                                                                                                                                               |        |                                                                                                                                    |
|                                                                                                                                                                                                                                                                                                                                                         |    |                                                                                                                                                                                                                                                                                                                                                                                                                               |        | Formatted: Justified Deleted: 21 Formatted: Page Number Formatted: Page Number                                                     |